# Cloud macro-physical properties in Saharan dust laden and dust free North Atlantic trade wind regimes: A lidar case study

Manuel Gutleben[1], Silke Groß[1], and Martin Wirth[1]

[1]Institut für Physik der Atmosphäre, Deutsches Zentrum für Luft- und Raumfahrt (DLR), 82234 Weßling, Germany.

**Correspondence:** Manuel Gutleben (manuel.gutleben@dlr.de)

**Abstract.** The Next-generation Aircraft Remote-sensing for VALidation studies (NARVAL) aimed at providing a better understanding of shallow marine trade wind clouds and their interplay with long-range transported elevated Saharan dust layers over the subtropical North Atlantic Ocean. Two airborne campaigns were conducted - the first one in December 2013 (winter) and the second one in in August 2016; the latter one during the peak season of transatlantic Saharan dust transport (summer). In this study airborne lidar measurements in the vicinity of Barbados performed during both campaigns are used to investigate possible differences between shallow marine cloud macro-physical properties in dust-free regions and regions comprising elevated Saharan dust layers as well as between different seasons. The cloud top height distribution derived in dust-laden regions differs from the one derived in dust-free regions and indicates that there are less and shallower clouds in the dust-laden than in dust-free trades. Additionally, a clear shift of the distribution to higher altitudes is observed in the dust-free winter season, compared to the summer-season. While during summer season most cloud tops are observed in heights ranging from $0.5$ to $1.0 \, \mathrm{km}$, most cloud tops in winter-season are detected between $2.0$ to $2.5 \, \mathrm{km}$. Moreover, it is found that regions comprising elevated Saharan dust layers show a larger fraction of small clouds and larger cloud free regions, compared to dust-free regions. The cloud fraction in the dust-laden summer trades is only $14 \, \%$ compared to a fraction of $31 \, \%$ and $37 \, \%$ in dust-free trades and the winter season. Dropsonde-measurements show that long-range transported Saharan dust layers come along with two additional inversions which counteract convective development, stabilize the stratification and may lead to a decrease of convection in those areas. Moreover, a decreasing trend of cloud fractions and cloud top heights with increasing dust layer vertical extent as well as aerosol optical depth is found.

## 1 Introduction

Saharan dust represents one of the main contributors to the atmospheres primary aerosol load. Huneeus et al. (2011) estimate that every year $400$ to $1000 \, \mathrm{Tg}$ of Saharan mineral dust are mobilized and transported over the North Atlantic Ocean within an elevated atmospheric layer: the so-called Saharan air layer (SAL; Carlson and Prospero (1972), Prospero and Carlson (1972)). Transatlantic Saharan dust transport shows its maximum during the northern hemispheric summer (Prospero and Lamb, 2003). In this period dust particles are frequently transported westwards and arrive in the Caribbean after approximately 5 days (transport speed: $\sim 1000 \, \mathrm{km \, d^{-1}}$ (Huang et al., 2010)). Sometimes Saharan dust is even transported as far as the coast of Mexico and Florida (Colarco, 2003; Wong et al., 2006).

During its long-range transport the SAL affects the Earth's radiation budget in two different ways. First, mineral dust aerosols may act as ice nucleating particles (INPs) or cloud condensation nuclei (CCN - only when being internally mixed with soluble material) in water and ice clouds, hence influencing cloud microphysics - this effect is referred to as 'indirect' mineral dust aerosol effect (Twomey, 1974, 1977; Karydis et al., 2011; Bègue et al., 2015; DeMott et al., 2015; Boose et al., 2016).

Thus, cloud formation, lifetime and occurrence as well as precipitation and ice formation may be manipulated by Saharan dust deposition into the cloud layer (Mahowald and Kiehl, 2003; Seifert et al., 2010). Secondly, dust particles absorb and scatter solar radiation during daytime and emit thermal radiation during nighttime. This so-called 'direct' mineral dust radiative effect modifies the atmospheric temperature profile and impacts the evolution of atmospheric stratification, sea surface temperature as well as cloud development (Carlson and Benjamin, 1980; Lau and Kim, 2007).

A large number of field campaigns aimed at getting a better understanding of the SAL as well as its interaction with clouds. The most extensive measurement series has probably been performed within the Saharan Mineral Dust Experiment series SAMUM-1 (Heintzenberg, 2009) and SAMUM-2 (Ansmann et al., 2011) followed by the Saharan Aerosol Long-range Transport and Aerosol-Cloud-Interaction Experiment SALTRACE (Weinzierl et al., 2017). Within this series of closure experiments, which included airborne and ground-based in-situ and remote sensing measurements as well as modeling efforts,

micro-physical, chemical and radiative properties of dust were investigated at the beginning of its long-range transport near the source regions as well as after its long-range transport in the vicinity of Barbados. Although the interaction of Saharan dust layers and clouds has already been a focus during these campaigns and other studies, e.g. by investigating glaciation of mixed-phase clouds (Ansmann et al., 2008; Seifert et al., 2010) or by exploring the relationship between shallow cumulus precipitation rates and radar measurements in dust-laden and dust-free environments (Lonitz et al., 2015), the impact of long-range

transported elevated Saharan dust on cloud macro-physical properties of subjacent trade wind clouds has not been studied.

Due to their occurrence at remote locations over the subtropical North Atlantic Ocean, it is difficult to study undisturbed trade wind cloud regimes and their interplay with Saharan air layers in the course of field campaigns with limited spatial coverage. Satellite measurements can of course provide information in these regions. Dunion and Velden (2004) used Geostationary Operational Environmental Satellite (GOES) infrared imagery to study the structural and dynamical characteristics of the SAL.

Wong and Dessler (2005) used MODIS (MODerate-resolution Imaging Spectroradiometer) satellite data to study the effect of the SAL on deep convection. Both studies found a suppressing effect of the SAL on deep convection and tropical cyclone activity. Wong and Dessler (2005) suggest that the convection barrier increases with SAL optical depth, especially over the eastern North Atlantic Ocean. They argue that the warmer and dryer air associated with the SAL rises the lifting condensation level as well as the level of free convection and therefore increases the energetic barrier to convection. These findings also sug-

gest a suppression of shallow marine cloud development due to long-range transported Saharan dust. Nevertheless, vertically resolved observations of suppressed marine cloudiness below long-range transported layers of Saharan dust over the Atlantic Ocean are missing so far.

Satellites with an active remote sensing payload, e.g. the Cloud-Aerosol Lidar and Infrared Pathfinder Satellite Observation (CALIPSO; (Winker et al., 2010)) and CloudSat (Stephens et al., 2002) provide vertically highly resolved measurements of

aerosol and cloud properties with nearly global coverage (Liu et al., 2008; Medeiros et al., 2010). Up to now, studies based on

active remote-sensing satellite data with focus on cloud macro-physical properties concentrated on long-term and large-scale observations, e.g. low-latitude boundary layer cloud cover (Medeiros et al., 2010), as the sensitivity of those instruments is too low to detect shallow marine clouds with high resolution. The upcoming EarthCARE (Earth Clouds, Aerosols and Radiation Explorer) satellite mission which is planned to be launched in 2021 (Illingworth et al., 2015) might change that in future due to its unique payload: a combination of lidar (Atmospheric Lidar - ATLID) and Cloud Profiling Radar (CPR). However, from spaceborne remote sensing in general it is hard to get an accurate aerosol retrieval during daylight conditions, which makes it difficult to study the interplay of SAL and clouds.

Besides satellite observations, measurements from long range research aircraft provide a valuable alternative to study the problem at hand. One such platform is the German High Altitude and LOng range research aircraft HALO (Krautstrunk and Giez, 2012). With HALO it is possible to perform measurements over both SAL-influenced and clear trade wind regions within the very same flight. During the NARVAL field campaigns (Next-generation Aircraft Remote-Sensing for VALidation Studies) HALO was used as a flying aerosol and cloud observatory (Stevens et al., 2019). For this purpose it was equipped with a combined active and passive remote sensing payload, including a radar and a lidar system. In addition, dropsondes were deployed to get information on the thermodynamic state of the atmosphere. The sensitivity of the radar system is not high enough to detect small-scale shallow cumulus clouds as well as aerosol layers. This is why this study only focuses on the retrieval of horizontal and vertical distributions of both aerosols and clouds from lidar measurements performed during the NARVAL field campaigns to study the impact of the SAL on subjacent marine cloud macro-physical properties (i.e. cloud fraction, cloud top height, cloud length).

Chapter 2 gives an overview of the NARVAL campaign series and a description of the employed lidar instrument. In Chapter 3 the general measurement situation during NARVAL is discussed and a detailed overview of the results is given. Moreover, the results are discussed and related to findings of other studies. A short summary along with the conclusion of this paper is presented in Chapter 4.

## 2 Instruments and methods

### 2.1 NARVAL

In December 2013 and August 2016 the Next-generation Aircraft Remote-Sensing for VALidation Studies (NARVAL; Klepp et al. (2014); Stevens et al. (2019)) were conducted to study the occurrence and formation of marine clouds during the subtropical North Atlantic dry and wet season. As Saharan dust transportation over the Atlantic Ocean occurs quite frequently in northern hemispheric summer months, measurements were also dedicated to investigate the influence of the SAL on underlying shallow trade wind clouds.

During both NARVAL-I-South (here for simplicity referred to as NARVAL-I) and NARVAL-II, HALO was operated eastward of Barbados. The aircraft has a maximum range of more than $12\,000\,\mathrm{km}$ and certified ceiling of $15.545\,\mathrm{km}$ altitude (max altitudes: NARVAL-I: $\sim14\,\mathrm{km}$; NARVAL-II: $\sim15\,\mathrm{km}$). During both campaigns it was equipped with a combined active and passive remote sensing payload including the lidar system WALES (Wirth et al., 2009), a $35.2\,\mathrm{GHz}$ cloud radar (Ewald et al.,

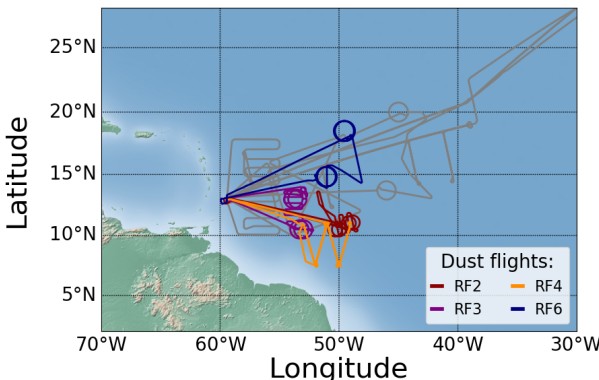

**Figure 1.** NARVAL research flight tracks: NARVAL-II dust-flights (color coded), NARVAL-I and NARVAL-II dust-free flights (grey).

**Table 1.** Overview of the conducted research flights during NARVAL-II in 2016 including dates, times of take-off and landing, total duration as well as research objectives and flight hours in SAL regions (all times given in UTC - note: Atlantic Standard Time = UTC-4; TBPB: Grantley Adams International Airport; EDMO: Airport Oberpfaffenhofen).

| Flight No. | Date | Take-off [UTC] | Landing [UTC] | Total duration | Research objective | Dust |
|:---:|:---:|:---:|:---:|:---:|:---:|:---:|
| RF1 | 08 Aug | 08:12 (EDMO) | 18:51 (TBPB) | 10:39 h | Transfer flight | - |
| RF2 | 10 Aug | 11:52 (TBPB) | 20:02 (TBPB) | 08:10 h | Dust/no dust flight and divergence | $\sim 2.3$ h |
| RF3 | 12 Aug | 11:43 (TBPB) | 19:37 (TBPB) | 07:54 h | Dust flight/Divergence | $\sim 6.5$ h |
| RF4 | 15 Aug | 11:47 (TBPB) | 19:46 (TBPB) | 07:59 h | Dust/no dust flight | $\sim 2.7$ h |
| RF5 | 17 Aug | 14:47 (TBPB) | 23:08 (TBPB) | 08:21 h | Satellite validation | - |
| RF6 | 19 Aug | 12:28 (TBPB) | 20:52 (TBPB) | 08:24 h | Dust/no dust flight and divergence | $\sim 4.5$ h |
| RF7 | 22 Aug | 13:16 (TBPB) | 20:57 (TBPB) | 07:41 h | ITCZ/Divergence | - |
| RF8 | 24 Aug | 12:43 (TBPB) | 20:55 (TBPB) | 08:12 h | Tropical storm Garcon/Divergence | - |
| RF9 | 26 Aug | 13:43 (TBPB) | 20:54 (TBPB) | 07:11 h | Tropical storm Garcon | - |
| RF10 | 30 Aug | 09:42 (TBPB) | 19:52 (EDMO) | 10:10 h | Transfer flight/Divergence | - |

2019), microwave radiometers (Mech et al., 2014), a hyper spectral imager (Ewald et al., 2016) and the Spectral Modular Airborne Radiation measurement System SMART instrument for radiation measurements (Wendisch et al., 2001). Additionally a large number of dropsondes were deployed to get information on the atmospheric state (NARVAL-I: 71; NARVAL-II: 218). From 8 to 30 August 2016, 10 research flights (RF) comprising a total of 85 flight hours were conducted (Figure 1). During

four of those flights, flight patterns were specifically designed for an investigation of Saharan air layers and their impact on subjacent marine trade wind cloud regimes. Moreover, studying the large scale atmospheric divergence was a main objective of the campaign (Bony and Stevens, 2019). This is why the flight patterns show many circles, i.e during RF2, 3, 6-8 and 10. Table 1 gives a detailed overview of all performed NARVAL-II research flights including the main research objectives.

This study focuses on the dust-laden research flights RF2, RF3, RF4 and RF6 of NARVAL-II. Cloud macrophysical properties measured during those flights are compared to properties observed during dust-free NARVAL-II flights. Data-sets obtained during the NARVAL-II transfer flights from and to Germany (i.e. RF1 and RF10) are not included in the analysis, because most measurements took place outside the trades and cirrus fields were present inside the trades. RF5 and RF7 are also excluded because cirrus fields covered most of the research area during RF5 and the objective of RF7 was to cross the Inter

Tropical Convergence Zone (ITCZ) for several times. NARVAL-I lidar measurements inside the trades (10 to 20° N) are used to compare obtained results from the 2016 summer season to the 2013 winter season.

In summary, 38 hours of measurements during the summer season (22 hours of lidar measurements during dust-free times, 16 hours of lidar measurements with SAL present) and 44 hours of measurements during the winter season are used to study differences in macro-physical cloud properties between the dust and non-dusty times and different seasons.

## 2.2   The WALES instrument

The WALES instrument (Wirth et al., 2009) is a combined airborne high spectral resolution (HSRL; Esselborn et al. (2008)) and water vapor differential absorption lidar system (DIAL), built and operated by the Institute for Atmospheric Physics of the German Aerospace Center (DLR). The system provides highly resolved information on the vertical distribution of water vapor mixing ratio from measurements at four wavelengths around $935\,nm$. Additionally, it is capable of polarization sensitive

measurements at $1064\,nm$ and $532\,nm$ wavelength. The $532\,nm$ channel is also equipped with High Spectral Resolution Lidar (HSRL) capability, which allows to determine the extinction coefficient without assumption on scattering properties of aerosol and cloud particles, hence enabling an enhanced characterization of them.

WALES measurements are performed in near nadir direction (2° - 3° off-nadir angle) and provide vertical profiles of particle backscatter, linear depolarization and extinction from the aircraft down to the ground level. The vertical resolution of the

WALES aerosol and cloud data is $15\,m$. The temporal resolution of the raw data is $5\,Hz$ and is averaged to $1\,Hz$ for a better signal-to noise ratio. This results in a horizontal resolution of approximately $200\,m$ at typical aircraft speed.

Depolarization data quality is ensured by frequent calibrations following to the ±45° method described by Freudenthaler et al. (2009). Remaining relative uncertainties in aerosol depolarization measurements are estimated to be in the range from $10\,\%$ to $16\,\%$ (Esselborn et al., 2008) and are primarily caused by atmospheric variations during the calibration. For backscatter and

extinction measurements relative uncertainties of less than $5\,\%$ and $10\,\%$ to $20\,\%$ have to be considered, respectively.

## 2.3   Dust layer detection

Based on the aerosol classification scheme described by Groß et al. (2013), WALES measurements can be used to identify and characterize layers of long-range transported Saharan dust. In this study the particle linear depolarization ratio at $532\,nm$

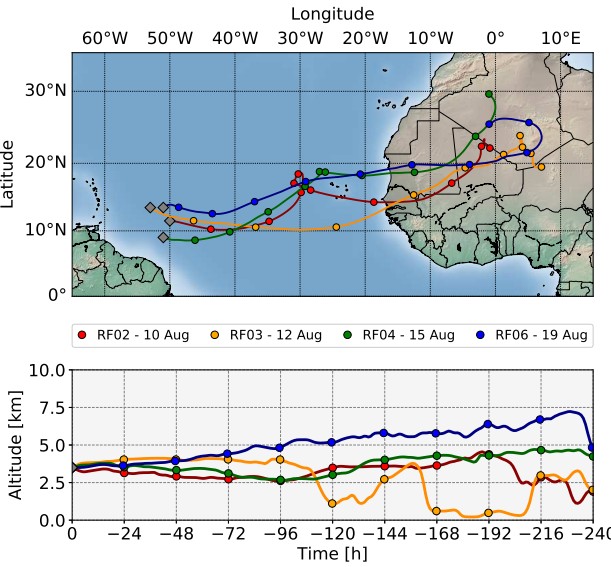

**Figure 2.** 10-day backward trajectories with starting points at the center of the respective Saharan air layers for the four NARVAL-II research flights leading over Saharan dust-laden trade wind regions (RF2, 3, 4 and 6).

($\delta_{p532}$) is used as an indicator for non-spherical dust particles. Saharan dust $\delta_{p532}$ near source regions was found to take values around $30\%$ (Freudenthaler et al., 2009; Tesche et al., 2009; Groß et al., 2011). This value does not change for long-range transported Saharan dust (Wiegner et al., 2011; Burton et al., 2015; Groß et al., 2015; Haarig et al., 2017). Thus $\delta_{p532}$ is a good proxy to distinguish long-range transported Saharan dust from less depolarizing marine boundary layer aerosols which typically take values around $3\%$ (Sakai et al., 2010; Burton et al., 2012; Groß et al., 2013). To reduce signal noise biases, an additional filter to flag mineral dust layers for regions with $532\,\mathrm{nm}$-backscatter ratios ($\mathrm{BSR}_{532}$) equal or higher $1.2$ is applied ($\mathrm{BSR}_{532} = 1 + \beta_{p532}/\beta_{m532}$ - where $\beta_{p532}$ and $\beta_{m532}$ are the particle and molecular backscatter coefficients). The origin of identified dust layers is further verified using calculated backward trajectories utilizing the HYbrid Single Particle Lagrangian Integrated Trajectory model (HYSPLIT model, Stein et al. (2015)) with NCEP GDAS (National Centers for Environmental Prediction Global Data Assimilation System) data input. Starting times and locations are chosen to match the center of detected mineral dust layers in the lidar profiles. The reliability of the backward trajectory calculations was checked by slightly modifying starting times and locations.

Once verified as transported Saharan dust layer, the WALES HSRL measurements are used to calculate the aerosol optical depth at $532\,\mathrm{nm}$ of both the detected Saharan dust layers ($\tau_{\mathrm{SAL}(532)}$) and the atmospheric column ranging from the aircraft down to ground level ($\tau_{\mathrm{tot}(532)}$). Additionally, the Saharan dust layer's vertical extent $\Delta z_{\mathrm{SAL}}$ is defined as the sum of all dust-flagged $15\,\mathrm{m}$-resolved height intervals within each vertical lidar profile.

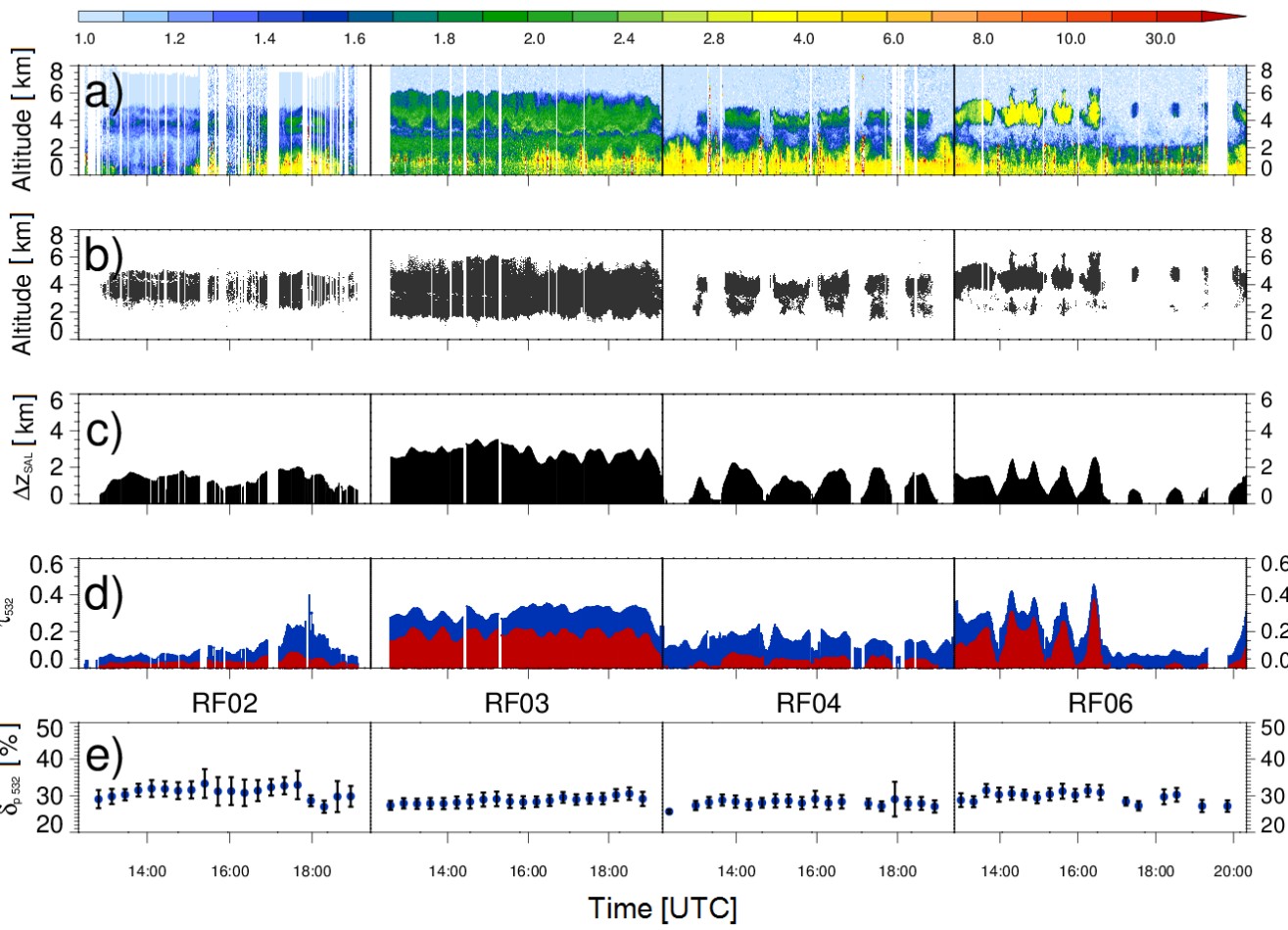

**Figure 3.** Overview of the four NARVAL-II research flights leading over Saharan dust-laden trade wind regions. (a) Cross sections of measured BSR at $532\,\text{nm}$ and (b) applied mineral dust mask. (c) 10-minute boxcar average of the calculated dust layer vertical extent $\Delta z_{\text{SAL}}$. (d) 10-minute boxcar average of the derived total dust aerosol optical depth from aircraft to ground level $\tau_{\text{tot}(532)}$ (blue) and aerosol layer optical depth $\tau_{\text{SAL}(532)}$ (red). (e) Mean values and standard deviations of the measured 10-minute averaged SAL particle linear depolarization ratio ($\delta_{\text{p}532}$).

## 2.4 Lidar-derived cloud macro-physical properties

Lidar derived cloud detection is usually performed using fixed signal thresholds (e.g. Medeiros et al. (2010); Nuijens et al. (2009, 2014)) or by applying wavelet covariance methods for the detection of sharp gradients to the backscattered signal (Gamage and Hagelberg, 1993). During NARVAL-II it was found that $BSR_{532}$ in the cloud-free marine trade wind boundary

layer as well as in the elevated SAL never exceeds a ratio of 10. Marine trade wind water-clouds are optically thick and thus take much larger values. Based on these findings and to avoid potential miscategorizations of sharp aerosol gradients as cloud tops using wavelet transforms a fixed threshold of $BSR_{532} = 20$ is used for the cloud/no-cloud decision.

To determine the cloud top height (CTH), the $BSR_{532}$ profile is scanned from flight level downwards to $250\,\mathrm{m}$ altitude and the first range bin where $BSR_{532}$ is greater or equal to the defined threshold is marked. Additionally, the whole profile is flagged as a 'cloud containing' profile. All 'cloud containing' profiles with cloud top heights in a certain altitude range are taken and divided by the total number of cloud-flagged profiles to obtain the CTH-fraction in the respective bin of the overall CTH-distribution. Similar to that the cloud fraction (CF) is defined as the number of all 'cloud containing' profiles divided by

the total number of vertical lidar profiles.

For the calculation of cloud lengths along the flight path neighboring cloud-flagged vertical profiles are connected. The cloud length is then determined as a function of the respective geolocations (aircraft latitude and longitude) and CTH using the haversine formula. Cloud gaps are calculated analogously by connecting neighboring cloud-free profiles. Due to the instruments maximum horizontal resolution of approximately $200\,\mathrm{m}$ it is possible to resolve minimum cloud (gap) lengths of $200\,\mathrm{m}$. It

should be mentioned that not the maximum cloud (gap) length of each individual cloud, but the along-track cloud (gap) length is derived. As a result, the amount of small clouds (gaps) in this study may be overestimated.

## 3    Results

### 3.1    Dust measurements during NARVAL-II

In the following the measurement situation during the four HALO-flights used to characterize long-range transported Saharan

dust layers (see Section 2.1) is summarized and their influence on subjacent marine trade wind clouds is investigated (Figure 3). The Saharan origin of the observed dust layers is verified using 10-day backward-trajectories with starting points at the center of the respective Saharan air layers (Figure 2). All observed dust layers traveled for 5 to $10\,\mathrm{days}$ from the Adrar-Hoggar-Aïr region in central Africa to the measurement location over the western North Atlantic Ocean. In central Africa the SAL is formed by intense surface heating and dry convection which mixes dust particles to altitudes of up to $6\,\mathrm{km}$ (Gamo, 1996).

During RF2 on 10 August a thin Saharan dust layer ($\Delta z_{SAL} < 2\,\mathrm{km}$) ranging from 2.5 to $5.0\,\mathrm{km}$ altitude was detected during the whole flight. A mean $\delta_{p532}$ of $30\,\%$ clearly classifies this elevated layer as a mineral dust layer. $\tau_{SAL(532)}$ took values around 0.15 - on average approximately $35\,\%$ of the total column aerosol optical depth during this RF. Unfortunately, bright and strongly reflecting clouds in the lidar field of view caused the safety circuit of the detector unit to shut down the device, causing some gaps in the continuous lidar data set.

In contrast to RF2, a vertically and optically thick dust layer was observed during the whole RF3 on 12 August. $\delta_{p532}$ of this layer ranged from $28\,\%$ to $30\,\%$, thus confirming the presence of Saharan mineral dust. The layer had a maximum vertical extent of $\sim 4\,\mathrm{km}$, showed aerosol optical depths around 0.2 and contributed on average with $60\,\%$ to the total column aerosol optical depth during that flight.

While RF2 and RF3 were designed for measurements solely in dust-laden regions, RF4 and RF6 on 15 and 19 August were

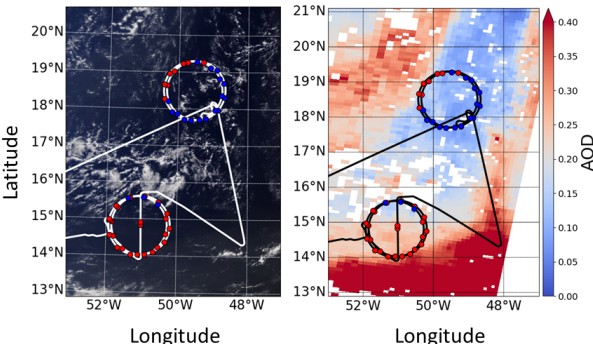

**Figure 4.** Flighttrack of RF6 on 19 Aug 2016 on top of the Terra-MODIS (MODerate-resolution Imaging Spectroradiometer) true color image (left) and the MODIS aerosol optical depth (AOD) product (right) at 13:40 UTC. Launched dropsondes are marked by colored dots (red dots: mineral dust laden regions, blue: dust free regions).

planned for measurements in both dust-laden and dust-free regions within the very same research flight. Flight tracks were chosen to cross dust-gradients frequently, resulting in multiple flight segments of dust and no dust along the flight track. Elevated aerosol layers showed mean $\delta_{p532}$ of $30\,\%$ and could therefore be identified as SAL. While the SAL during RF4 ranged on average from 2.5 to $4.5\,\mathrm{km}$, it reached higher to almost $6\,\mathrm{km}$ altitude during RF5. With $\tau_{SAL(532)} \approx 0.1$ the dust layer during RF4 contributed on average $25\,\%$ to $\tau_{tot(532)}$. $\tau_{SAL(532)}$ during RF6 took higher values of up to $0.4$ and showed a mean contribution of $51\,\%$ to $\tau_{tot(532)}$.

The following case study presents a detailed description of RF6 including an analysis of dropsonde-profiles in dust laden and dust free regions.

### 3.2   Case study - 19 Aug 2016

RF6 on 19 August 2016 took place in the area between $48°\mathrm{W}$ to $60°\mathrm{W}$ and $13°\mathrm{N}$ to $19°\mathrm{N}$ (Figure 4). The Intertropical Convergence Zone (ITCZ) and associated deep convection were located $\sim$$550\,\mathrm{km}$ south of the flight track at around $10°\mathrm{N}$ and it is not expected to have an influence of the ITCZ on our analysis. RF6 was planned to cross a sharp gradient between a dust-laden and a clear region in an altitude of approximately $8.25\,\mathrm{km}$ with about one half of the measurement time in dust-laden and the other half in dust-free regions. The circular patterns of the flight track were flown for dropsonde-based divergence measurements. Whereas the first pair of circles was performed over a heavily dust-laden region in the southern part of the flight track, the second pair was performed in the northern part over an almost dust-free region. This is also seen in MODIS aerosol optical depth imagery at 13:40 UTC in Figure 4 (right) where the region around the southern circle shows a maximum aerosol optical depth greater $0.4$.

Measured cross sections of $BSR_{532}$ and the derived mineral-dust mask (Figure 3 (a, RF6) and (b, RF6)) show pronounced

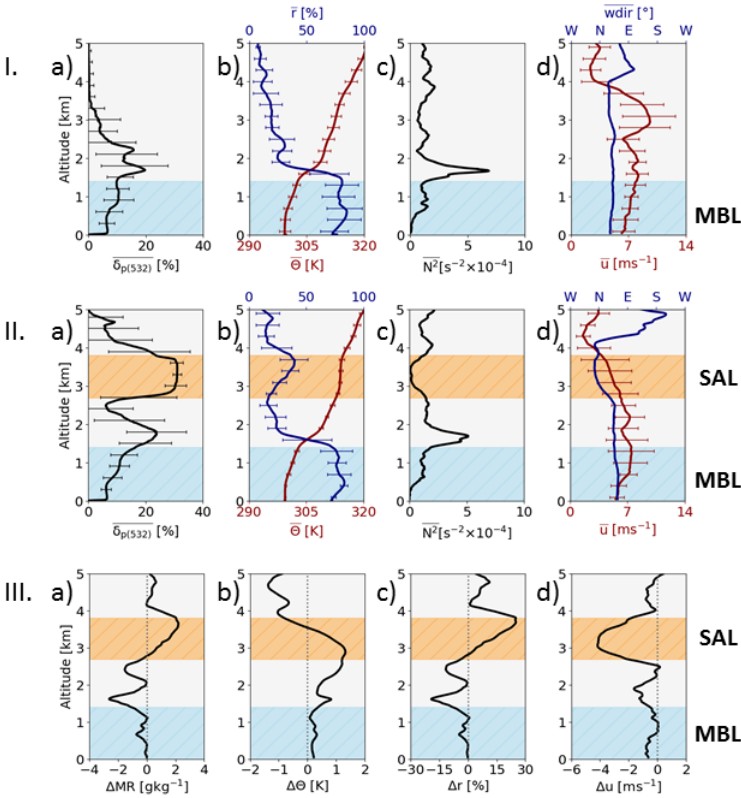

**Figure 5.** Mean vertical WALES lidar profiles of $\delta_{p532}$ and mean vertical dropsonde profiles of relative humidity ($r$), potential temperature ($\theta$), squared Brunt-Väisälä frequency $N^2 = \frac{g}{\Theta}\frac{d\Theta}{dz}$ as well as wind speed (u) and direction (wdir) in dust-free (I.(a-d)) and dust-laden (II.(a-d)) regions during RF6 on 19 Aug 2016 (horizontal bars indicate standard deviations). III.(a-d): Differences in water vapor mass mixing ratio (MR), potential temperature, relative humidity and wind speed between the two regions. Shaded regions mark the Marine Boundary Layer (MBL, blue) and the Saharan Air Layer (SAL, orange).

elevated mineral dust layers ranging from 2.5 to 5.0 km altitude, horizontally alternating with dust-free profile regions. Due to the conducted divergence measurements, dropsondes were launched frequently along the circular flight tracks and are used to compare vertical profiles of meteorological parameters in dust-laden to those in dust-free regions. For this purpose mean profiles of potential temperature $\theta$, relative humidity and water vapor mass mixing ratio ($r$, MR) as well as wind speed and direction (u, wdir) of all dropsonde-measurements in the respective dust-laden and dust-free regions are compared in Figure 5 (I.(b-d), II.(b-d) and III.(a-d)). Additionally, lidar-derived $\delta_{p532}$ is analyzed for both regimes (Figure 5, I.a and II.a).

Inside the SAL-region a three-layer structure is present: 1) the marine boundary layer (MBL), reaching up to approximately 1.3 km height. $\delta_{p532}$ smaller 10 % indicates that marine aerosols are the dominant contributor to the aerosol composition of the MBL; 2) a transition or mixed layer extending from the MBL-top to ∼2.8 km altitude with varying values of $\delta_{p532}$

$(10\% < \delta_{p532} < 20\%)$; and 3) the elevated SAL, with typical $\delta_{p532}$ for long-range transported Saharan dust ($\delta_{p532} \sim 30\%$) ranging from $2.8$ to $3.8\,\mathrm{km}$ height.

The mean dust-free $\delta_{p532}$-profile of the MBL and transition layer looks quite similar to mean dust-laden $\delta_{p532}$-profile. However, no SAL-signature is detected. The low CF in the dust-laden region (southern part of the flight track) which is visible in the MODIS image (Figure 4) is also evident in lidar measurements after the application of the described threshold method for cloud detection. Whereas a lot of cloud tops in heights ranging from $0.5$ to $1.5\,\mathrm{km}$ altitude are detected in the northern part of the flight track (after about 16:45 UTC), almost no cloud is detected along the earlier southern flight path - with the exception

of the transition region to the dust free area (cloud top heights at $\sim 1.5\,\mathrm{km}$ altitude).

This is also evident in the calculated cloud fractions. CF is $20\%$ in dust-free regions. In the SAL-region however, CF decreases to $11\%$ (including the clouds developing at the edges of the dust layer). Another characteristic of clouds in SAL-regions is that their CTH is rarely higher than approximately $1\,\mathrm{km}$. However, in dust free regions cloud top heights reach almost twice as high and up to $2\,\mathrm{km}$. Divergence measurements discussed by Bony and Stevens (2019) and Stevens et al. (2019) show that

dynamical properties in the two regions are different as well. They found that MBL-vertical velocity in the dust-free regime is directed upwards and could explain the observed increased cloud top heights.

For an investigation of the question why vertical wind speeds, cloud tops and cloud fractions are higher in the dust-free regime than in the SAL-regime differences in meteorological parameters between SAL-regions and dust-free regions are analyzed by discussing mean profiles of all dropsonde measurements in the respective regions. Both the dust-laden and the dust-free

region clearly indicate the so-called trade wind inversion (TWI) in an altitude range from $1.5$ to $1.8\,\mathrm{km}$ height capping the moist MBL. The TWI is characterized by a rapid temperature decrease of about $4\,\mathrm{K}$ within $400\,\mathrm{m}$ (not shown) and a strong hydrolapse (relative humidity (r) drops from $>80\%$ to $\sim 30\%$). In both regimes the MBL itself can be divided into a sub-cloud layer which extends from the ocean surface to $0.5$ to $0.7\,\mathrm{km}$ and a cloud layer (Groß et al., 2016) which extends from the sub-cloud layer top to the TWI ($\sim 0.5$ to $\sim 1.8\,\mathrm{km}$). Those two regions can be identified in profiles of $\Theta$ and humidity. Whereas

the sub-cloud layer is well mixed ($\Theta = \mathrm{const.}$, $\mathrm{MR} = \mathrm{const.}$), the cloud layer shows a conditionally unstable lapse rate of $5$ to $7\,\mathrm{K\,km^{-1}}$ (saturated air parcels are unstable to vertical displacement). Overall, measurements of $\Theta$ and humidity show a stronger variation in the dust-free MBL than in the dust laden one, suggesting the presence of more boundary layer clouds in dust-free regions.

Nuijens et al. (2009) and Nuijens and Stevens (2012) found that high wind speeds near surface correspond to an increase

of boundary layer humidity leading to a deepening of the cloud layer and increased area rainfall. Lonitz et al. (2015) used Large Eddy Simulations to show how higher relative humidities associated with observed dusty boundary layers changes the evolution of the cloud layer. However, when comparing boundary layer wind speed and humidity in the two regimes no distinct differences can be observed, indicating that some other mechanism must be responsible for the observed differences in vertical wind speed, cloud fraction and cloud top height. The MBL of both regimes is dominated by north-easterly winds with speeds

around $7\,\mathrm{m\,s^{-1}}$. In dust-laden regions wind speeds in SAL-altitudes are by $4\,\mathrm{m\,s^{-1}}$ lower than in the dust-free regions. This suggests that the SAL represents a decoupled layer which penetrates into the trade-wind regime. Moreover, enhanced amounts of water vapor are observed inside the long-range transported SAL. Relative humidity and water vapor mass mixing ratio show

an increase of $2\,\mathrm{g\,kg^{-1}}$ in SAL regions compared to the dust-free trade wind region. Such an increase has already been observed by Jung et al. (2013). From radiosonde measurements they found that the SAL transports moisture from Africa towards the

Caribbean and gets moistened during transport by upwelling surface fluxes.

For a better visualization of atmospheric stability the squared Brunt Väisälä frequency $N^2 = \frac{g}{\Theta}\frac{d\Theta}{dz}$, with $g$ being the gravity of the Earth, is shown. $N^2$ shows regions of high atmospheric stability and thus strong restoring forces for a vertical air parcel displacement at the inversion altitudes. Enhanced atmospheric stability is found at the TWI for both regimes. At higher altitudes $N^2$-profiles look different. In dust-laden regions the lower and upper boundary of the SAL are characterized by two additional

well-known inversions (Carlson and Prospero, 1972; Dunion and Velden, 2004; Ismail et al., 2010). Inside the layer $N^2$ is almost zero - indicating a well mixed SAL-regime. Furthermore, $\Theta$ points towards a neutral stratification in the interior of the SAL since it does not change with altitude. Altogether, a total of three prominent inversion layers counteract convective development in dust-laden regions, whereas in dust-free regions just the trade wind inversion is present.

In conclusion, one can suggest that the SAL potentially modifies radiative transfer, atmospheric stability as well as the evolution

of vertical velocities, hence representing a proxy for reduced amounts of clouds and lower cloud top heights. To discuss this hypothesis, differences in cloud macro-physical properties are investigated for the whole NARVAL field campaign series.

## 3.3 Differences in cloud macro-physical properties

### 3.3.1 Cloud fraction and cloud top height

A first indicator for differences in marine trade wind cloud occurrence is the cloud fraction CF. During NARVAL-II a total num-

ber of $3.2 \times 10^4$ one second resolved cloud tops were detected in trade wind regions ($N_{CT(dust)} = 8 \times 10^3$; $N_{CT(nodust)} = 2.4 \times 10^4$). They contribute to an overall observed CF of $24\,\%$ within the measurement period. In dust-free regions a CF of $31\,\%$ was derived, while in SAL-regions CF was smaller by a factor of more than two ($14\,\%$). In winter season (NARVAL-I) an almost three times higher CF of $37\,\%$ is derived. The next parameter to look for differences between the three regions is the CTH-distribution (Figure 6). In the SAL-regions only a small fraction of clouds exceeds an altitude of $2\,\mathrm{km}$ and no cloud top is found

at altitudes greater $2.5\,\mathrm{km}$. The majority of cloud top heights ($\sim 61\,\%$) is found within the altitude range from $0.5$ to $1.0\,\mathrm{km}$ in lower altitudes of the MBL-cloud layer. $26\,\%$ of all detected cloud top heights are located in the $1.0$ to $1.5\,\mathrm{km}$ height interval and only $11\,\%$ of that fraction contribute to the interval from $1.5$ to $2.0\,\mathrm{km}$ altitude.

Cloud tops in altitudes $>2.5\,\mathrm{km}$ including deeper reaching convection with maximum top heights of $6\,\mathrm{km}$ are found in $\sim 16\,\%$ of all dust-free cloud profiles. Below around $3\,\mathrm{km}$ altitude the CTH-distribution shows a two-modal structure with two local

maxima ranging from $0.5$ to $1.0\,\mathrm{km}$ ($\sim 35\,\%$) and $1.5$ to $2.0\,\mathrm{km}$ altitude ($\sim 20\,\%$). Several clouds were also detected in the lowermost $0.5\,\mathrm{km}$ of the atmosphere ($\sim 1\,\%$). Most likely those clouds are evolving or dissipating clouds at the bottom of the cloud layer.

In the dust-free winter season a shift of the distribution to higher altitudes is observed, since most cloud tops were sampled in the interval from $2$ to $2.5\,\mathrm{km}$ altitude ($\sim 39\,\%$). However, no cloud was observed in altitudes greater $3.5\,\mathrm{km}$. This shift is caused by a slightly higher TWI in winter months, shown in Stevens et al. (2017) who compare mean dropsonde-profiles of

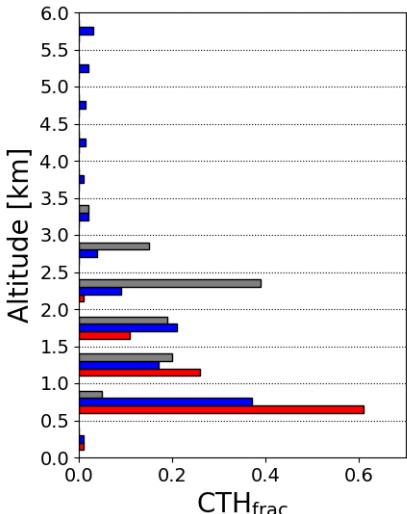

**Figure 6.** Histograms of detected cloud top height fractions during NARVAL-I and II with bins of $0.5\,\text{km}$ size. Red bars illustrate the distribution of cloud top height fractions in SAL-regions. Blue bars represent the derived cloud top height distribution from measurements in the dust-free trades during NARVAL-II. Grey bars show the derived cloud top height in the dust-free winter season during NARVAL-I.

water vapor mixing ratio during in NARVAL-I and II.

The statistical significance of observed differences in the distributions was checked by randomly resampling the respective
5    data-sets to smaller sub-sets and by comparing the shapes of the resulting distributions to the shape of the overall distributions.
The shapes of the resampled distributions showed no major differences compared to the overall distributions, thus it can be
concluded that our NARVAL-II measurements indicate the presence of less and shallower clouds in Saharan dust laden trade
wind regions compared to dust-free regions.

### 3.3.2    Cloud lengths and cloud gaps

10   As next step the cloud length and cloud gap length distributions of marine trade wind clouds in SAL-regions and mineral dust
free regions are investigated (Figure 7, top). A total of 3688 and 2355 clouds were observed in dust free and dust-laden regions
during NARVAL-II and 5010 clouds were detected during NARVAL-I in dust-free winter. In all three samples clouds with a
horizontal extent of less than $0.5\,\text{km}$ are by far the most prominent ones. Whereas $72\,\%$ of all clouds in SAL-regions are of this
length, $65\,\%$ of clouds detected in clear summer trade wind regions and $61\,\%$ in winter time measurements contribute to this
15   length-interval. Both regions show a decreasing trend in frequency of cloud length occurrence for lengths of up to $5\,\text{km}$. Rela-
tive frequency drops to $\sim 17\,\%$ (dust-laden), $16\,\%$ (dust-free) and $16\,\%$ (NARVAL-I) in the length interval from $0.5$ to $1.0\,\text{km}$.
Only $5\,\%$ of all clouds in dusty regions are observed to have a horizontal extent greater than $2\,\text{km}$. This fraction almost doubles
to $9\,\%$ in dust-free regions and in winter months. The main contributor to this fraction are clouds with horizontal extents of

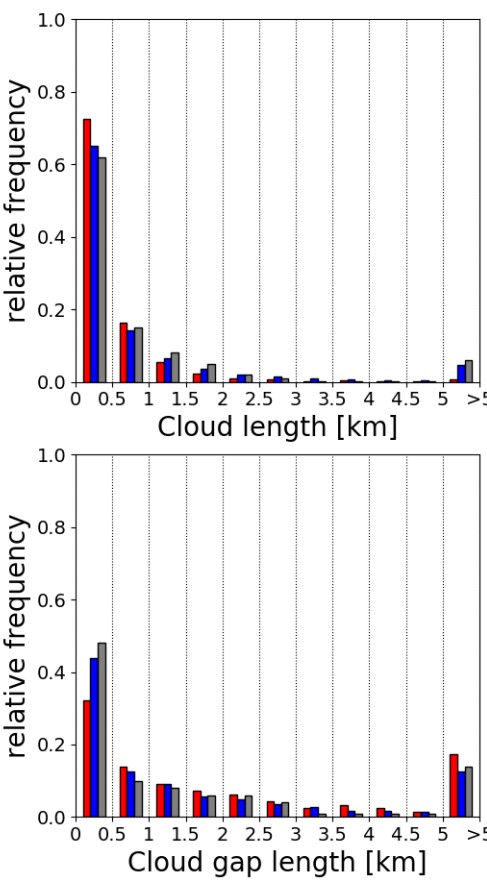

**Figure 7.** Histograms of detected cloud lengths (top) and cloud gap lengths (bottom). Red bars illustrate the distribution of marine low cloud (gap) lengths located below Saharan dust layers. Blue bars represent the distribution derived from measurements in the dust-free trades during NARVAL-II. Grey bars show the derived distribution in the dust-free winter season during NARVAL-I.

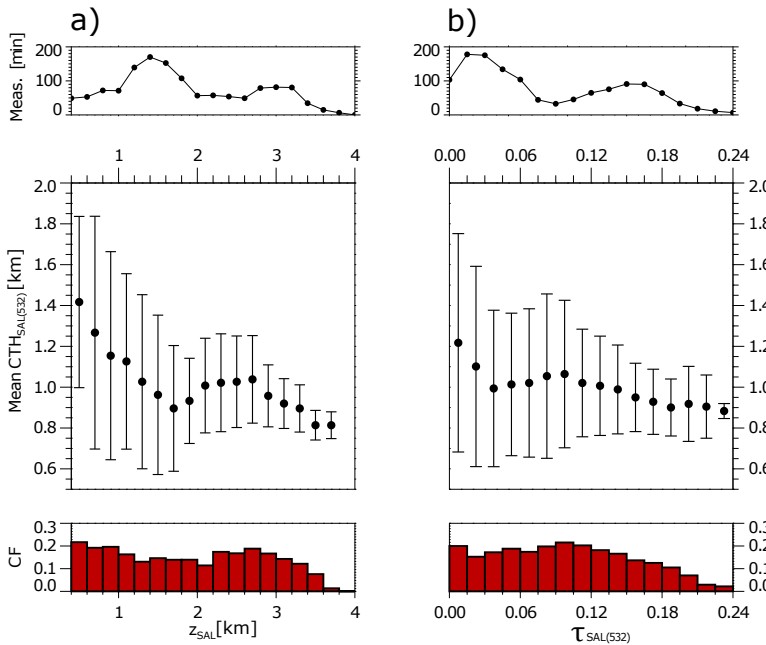

**Figure 8.** (a) Mean cloud top heights (middle) and cloud fraction (bottom) of clouds detected below Saharan dust layers as a function of Saharan dust layer vertical extent ($\Delta z_{\mathrm{SAL}}$) - bin-interval: $0.2\,\mathrm{km}$, (b) Mean cloud top heights (middle) and cloud fraction (bottom) of clouds detected below Saharan dust layers as a function of Saharan dust layer optical depth ($\tau_{\mathrm{SAL}(532)}$) at $532\,\mathrm{nm}$ wavelength - bin-interval: $0.015$. Bars mark respective standard deviations of mean cloud top heights ($1\sigma$). The uppermost graphs in (a) and (b) illustrate summed measurement-times in each interval.

more than $5\,\mathrm{km}$ ($4$ and $5\,\%$). Clouds of this length are basically only found outside dust-laden regions.

Another important parameter to highlight differences of cloudiness between SAL-regions and dust-free regions is the cloud
5  gap length (Figure 7, bottom). Similar to the distribution of cloud lengths, also cloud gap frequencies decrease with increasing cloud gap length. In all three regimes cloud gaps shorter $0.5\,\mathrm{km}$ are dominating. They contribute with $45\,\%$ and $35\,\%$ to the total amount of observed cloud gaps in dust-free and dust-laden regions during NARVAL-II and with even $48\,\%$ in winter months. A different picture emerges, when looking at the amount of cloud gaps greater than $5\,\mathrm{km}$. A fraction of $17\,\%$ is found to be greater than $5\,\mathrm{km}$ below dust layers, whereas in dust-free regions and winter months these gap sizes contribute with $12\,\%$
10  and $14\,\%$ to the distribution. Cloud gap fractions in range-bins from $1.5$ to $4.5\,\mathrm{km}$ decrease in both regions consistently with increasing cloud gap length.

The significance of the distribution-properties was again double-checked by the comparison to randomly resampled sub-datasets. Overall, the cloud length and gap length distributions (Figure 7) indicate, that the dust-laden trade wind regimes

during NARVAL-II were characterized by a larger amount of small scale clouds and slightly greater cloud gaps, compared to the dust free and winter regimes.

### 3.3.3   Connecting dust and cloud properties

As a further step the observed CTH and CF are related to the geometrical and optical depth ($\Delta z_{SAL}$ and $\tau_{SAL(532)}$) of overlying mineral dust layers (Figure 8). Cloud fractions and heights in dust-flagged profiles of all four research flights are grouped together with respect to similar $\Delta z_{SAL}$ (bin width: $0.2\,\mathrm{km}$) and $\tau_{SAL(532)}$ (bin width: $0.015$). During NARVAL-II Saharan dust layers with maximum vertical extents of $4\,\mathrm{km}$ and maximum optical depths of $0.4$ were observed (Figure 3). However, below optically thick dust layers ($\tau_{SAL(532)} > 0.24$; $z_{SAL} > 3.8\,\mathrm{km}$) not any cloud has been detected.

The distribution of CTH as a function of $\Delta z_{SAL}$ shows that up to a layer thickness of $1.8\,\mathrm{km}$ mean CTH decreases with increasing $\Delta z_{SAL}$ from $\sim$1.4 to $\sim$0.8 km altitude. For a greater layer thickness ($\Delta z_{SAL} > 1.8\,\mathrm{km}$) this trend is not evident anymore. A further increase in $\Delta z_{SAL}$ does not imply a significant decrease in mean CTH - in some bin-intervals the mean CTH even increases slightly. Mean cloud top heights vary strongly below vertically thin dust layers ($\sigma = 0.5\,\mathrm{km}$) - an indication for the presence of both shallow developing convective clouds and higher reaching trade wind clouds within the MBL. With

increasing $\Delta z_{SAL}$ the variability of mean CTH decreases and reduces to $\sigma < 0.2\,\mathrm{km}$ for $\Delta z_{SAL} < 3\,\mathrm{km}$. This suggests that the cloud layer indeed lowers and that the few evolving clouds are confined to low levels of the MBL. The CF distribution as a function of $\Delta z_{SAL}$ does not show any distinct trend for geometrically thin layers. For $\Delta z_{SAL} < 1.0\,\mathrm{km}$ CF takes values around $20\,\%$ - only slightly lower values than the CF derived from measurements in dust-free regions. For vertical extents ranging from $1.0\,\mathrm{km} < \Delta z_{SAL} < 2.6\,\mathrm{km}$ no clear decrease in CF is detected. In this range CF varies around $15\,\%$ and even increases

slightly. A clear decreasing trend of CF with increasing $\Delta z_{SAL}$ is obvious only for $\Delta z_{SAL} > 2.6\,\mathrm{km}$.

Next, the CTH distribution as a function of dust layer optical depth $\tau_{SAL(532)}$ is analyzed. Up to a value of $\tau_{SAL(532)} \sim 0.05$ the mean CTH decreases with increasing optical depth of the aerosol layer. The mean CTH drops from $\sim 1.3\,\mathrm{km}$ to $\sim 1.0\,\mathrm{km}$ in this region. A further increase of $\tau_{SAL(532)}$ to a value of about $0.12$ does not show any further decrease in mean CTH. This is in line with the observed decrease in CF as a function of dust layer optical depth in this range. The observed CF increases slightly

from $15\,\%$ to $20\,\%$ for small SAL-optical depths ($\tau_{SAL(532)} < 0.12$). At the upper tail of the distribution ($0.12 < \tau_{SAL(532)}$) the mean CTH as well as the CF decrease again. CF shows a steady decrease of about $20\,\%$ in the range from $\tau_{SAL(532)} = 0.12$ to $0.24$. Moreover, the variability of mean CTH in that range gets smaller, again indicating that higher-reaching convection is suppressed.

For the interpretation of these distributions the accumulated measurement-time in the respective intervals as well as the con-

tribution of different research flights have to be taken into account. Mainly data collected in the course of RF3 contributes to SAL-measurements in the ranges $0.09 < \tau_{SAL(532)} < 0.24$ and $2\,\mathrm{km} < \Delta z_{SAL} < 4\,\mathrm{km}$ (Figure 3), thus being the main contributor to observed increases of mean CTH and CF in regions of high $\tau_{SAL(532)}$ and $\Delta z_{SAL}$. The remaining research flights (RF2, RF4 and RF6), were characterized by thinner dust layers that were rather decoupled from the MBL and contribute to regions of small $\tau_{SAL(532)}$ and $\Delta z_{SAL}$.

Altogether, a decreasing trend of CTH and CF as a function of dust layer optical depth and vertical extent was detected

during research flights over elevated and long-range transported Saharan dust layers. However, RF3 showed a predominant and strongly pronounced transition layer that possibly altered the cloud layer resulting in an increased CF and CTH in the respective intervals of $\tau_{\mathrm{SAL(532)}}$ and $\Delta z_{\mathrm{SAL}}$.

## 4  Summary and Conclusion

In this study airborne lidar measurements performed on-board the German high altitude and long-range research aircraft HALO during the NARVAL experiments over the North Atlantic trade wind region were used to investigate whether marine low cloud macro-physical properties change in the presence of overlying long-range transported Saharan dust layers. Significant differences in the CTH distribution as well as in the cloud length and cloud gap length distribution were found for flights in SAL-regions compared to the distributions derived from flights in dust-free regions. It can be summarized that during times with Saharan air layers less, shallower and smaller clouds are present than during times without Saharan air layers. The overall derived cloud fraction in the dust-laden trade wind summer regime is $14\,\%$ and thus a factor of two smaller than the cloud fraction of $31\,\%$ and $37\,\%$ derived from observations in the dust-free regime and the winter season. These results are in good agreement with results of previous satellite remote sensing studies (Dunion and Velden, 2004) and model studies (Wong and Dessler, 2005; Stephens et al., 2004) which also suggest a convection-suppressing characteristic of the SAL. Some of those studies suggest that the main player of the suppression characteristic is a dry anomaly in SAL-altitudes. However, all observed long-range transported Saharan air layers during NARVAL-II were not found to come along with dry anomalies, but were rather showing enhanced humidities (compared to the surrounding dry free trade wind atmosphere) in the range from 2 to $4\,\mathrm{g\,kg^{-1}}$. Saharan air layers frequently show water vapor mixing ratios in this range over Africa (Marsham et al., 2008). During the transport towards the Caribbean the SAL conserves the received moisture and takes up additional one from upwelling surface fluxes during transport (Jung et al., 2013). Nevertheless, a suppressing characteristic of the SAL on subjacent marine clouds is evident as well.

Wong and Dessler (2005) also showed that the convection barrier increases with SAL-aerosol optical depth. To investigate a possible relation between SAL optical depth or layer vertical extent and marine trade wind CTH, the CTH and CF-distribution was analyzed as a function of SAL vertical extent and optical depth. It was found that mean CTH decreases with increasing layer vertical extent for vertically thin layers ($<1.5\,\mathrm{km}$). Additionally, the mean CTH-variability for these layers is high, indicating the occurrence of higher-reaching clouds in those regions. There is no significant decrease of mean CTH for thicker dust layers, but a reduction of CTH-variability could be derived. Also a decrease in mean CTH-variability with increasing dust layer optical thickness starting at $\tau_{\mathrm{SAL(532)}} \approx 1.2$ could be detected. Moreover, a decrease in CF comes along with this reduction in variability of the mean CTH. Below optically thick dust layers with $\tau_{\mathrm{SAL(532)}} > 0.24$ not any cloud was detected. These results indicate that optically and vertically thick elevated Saharan dust layers have a greater suppressing effect on convection below than optically and vertically thin layers. Dropsonde profiles of potential temperature $\theta$ and the squared Brunt-Väisälä frequency $N^2$ in dust-laden trade wind regions indicate two inversions at the bottom and the top of the SAL which additionally counteract convective development. Those two SAL-related additional inversion layers are an explanation

why there are less and shallower clouds in SAL regions and why thick and pronounced dust layers introduce a more stable stratification to the trade wind regime than less pronounced ones.

Altogether, the NARVAL lidar measurements indicate that there is a strong correlation between the presence of elevated and long-range transported Saharan dust layers and the occurrence and macro-physical properties of subjacent marine low clouds. It is shown that Saharan dust can be used as a proxy for a decrease in subjacent trade wind cloud length and cloud top height. Further reaching questions regarding changes in radiation caused by the dust layer and its moisture, changes in the general circulation patterns or the settling of dust particles into the cloud layer (Groß et al., 2016) could not be addressed within the present work and are left to future studies and field campaigns, e.g. the upcoming $EUREC^4A$ field campaign (ElUcidating the RolE of Clouds-Circulation Coupling in ClimAte) in early 2020 (Bony et al., 2017).

*Author contributions.* In the framework of the NARVAL-II field experiment Martin Wirth and Silke Groß contributed to carry out all airborne lidar measurements used in this study. Martin Wirth did the initial data processing. Manuel Gutleben performed all analytic computations, statistically analyzed the data set and took the lead in writing the manuscript under consultation of Silke Groß. All authors discussed the results and contributed to the final manuscript.

*Competing interests.* none declared

*Acknowledgements.* The authors like to thank the staff members of the DLR HALO aircraft from DLR Flight Experiments for preparing and performing the measurement flights. The data used in this publication was collected during the NARVAL (Next-generation Aircraft Remote-sensing for VALidation Studies) campaign series and is made available through the DLR Institute for Atmospheric Physics. Moreover, the authors gratefully acknowledge all research scientists who helped to launch the dropsondes and the two anonymous referees who helped to improve this study. NARVAL was funded with support of the Max Planck Society, the German Research Foundation (DFG, Priority Program: HALO-SSP 1294) and the German Aerospace Center (DLR). This study was financed by a DLR VO-R young investigator group within the Institute of Atmospheric Physics.

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
