# Peer review of "Cloud macro-physical properties in Saharan dust laden and dust free North Atlantic trade wind regimes: A lidar case study"

_Atmospheric Chemistry and Physics, 2019_

## Referee Comment (RC1) · Anonymous Referee #1 · 19 Mar 2019

General comments: This paper investigates the link between the Saharan aerosol layer and cloudiness using lidar measurements and dropsondes data of few selected NAR-VAL flights. They find that situations with SAL correlate with lower cloud top heights and a larger fraction of small clouds. The study is very detailed in presenting numbers and doing appropriate statistical testing. Reading through the paper suggests that the aerosols are causing the differences in cloudiness but how can an elevated dust layer imprint itself on the lower laying clouds? Furthermore, what is not quite clear to me is how the meteorology during periods of SAL differs to periods without dust and how this might influence the cloudiness. Other studies (e.g. Lonitz et. al 2015) have shown, that e.g. slight differences in relative humidity can also cause differences in trade wind

clouds to a comparable level as differences in aerosol load can do. Also effects of wind should be discussed, as done by Nuijens and Stevens in 2012 using Large Eddy Simulations. Having access to meteorological data from the dropsondes, I would like to see that the differences in relative humidity and wind are analysed in greater depth (not just shown as done in Fig. 4 for humidity) and a discussion about if dust or the meteorology associated with dust could potentially can cause the observed changes in cloudiness.

Specific comments:

Introduction: Please elaborate in more detail how dust and the formation of cloud interact, how the aerosol could alter clouds and why dust can act as a good CCN.

p2, l18/19 is not true. Lonitz et al 2015 (https://journals.ametsoc.org/doi/10.1175/JAS-D-14-0348.1) have studied this.

Section 2.4: Do you also check if the neighbouring cloudy profiles have the cloud in similar height levels? If not, this might have an impact on the results shown in section 3.3.

Figure 4 and p15, l 5-7: Was the relative humidity measured by the dropsondes or derived by the lidar? How does dry Saharan air layers relate to an increase in humidity?

P15, l14/15: How does the suppression work. Please elaborate

Technical comments:

Often the word "underneath" is used instead of "below", e.g. p2 l8, caption of figure 7

P2, l35: Citation incorrect: Stevens et al 2019.

P10, l19: "first" instead of "fist"

Sec 3.3.2: How do you derive the cloud length in kilometers? My guess is that you assume some relationship between the speed of the aircraft and time?!

---

## Referee Comment (RC2) · Anonymous Referee #2 · 20 Mar 2019

General comments:

The manuscript "Cloud macro-physical properties in Saharan dust laden and dust free North Atlantic trade wind regimes: A lidar study" is focusing on measurements from the NARVAL-II campaign, which took place in the tropical Atlantic trade wind region. The authors use aircraft based lidar measurements to compare Saharan dust laden regions with dust-free regions and look at the influence on the macro-physical cloud properties. The main findings in the manuscript are some significant differences in cloud top height, cloud length and cloud gap length distributions between dust laden and dust-free regions. All in all, the manuscript is very well written. I would suggest

the manuscript to be published after major revision. This should address the following points:

Major comments:

Concerning the structure: The weakest part of this paper is the missing explanation about how the dust in the higher atmosphere is influencing the cloud macro-physical properties. The authors mention some papers in the introduction and compare their results to some publications in the conclusions, but there is a section missing which explains the results in detail. For example, the authors mention a paper which shows a convection-suppressing characteristic of the SAL with the main player being a dry anomaly in SAL-altitudes, but they cannot confirm a dry anomaly in SAL-altitudes with their measurements. For this reason, the authors should make clear why they still believe that SAL causes a suppressing characteristic on the clouds. In addition, the authors should make it very clear in the beginning (probably in the abstract or in the headline) that this study is a case study, which is based on four research flights.

Concerning the dataset: In this study, the authors focus on a dataset from the NARVAL-II campaign. In particular, they focus on four research flights where dust and dust free regions were sampled. If the authors would also look at the measurements from the first NARVAL campaign in 2013, they could extend their dataset and improve the statistics.

Concerning the analysis: The authors compare the dust layer properties with the macro- physical properties of the clouds and see some correlations. However, the authors don't exclude other possibilities, which could cause the changes in clouds as well. For example, in their case study from the 19. August 2016, the authors compare a dust-laden region with a dust-free region and argue that in the latter, the cloud top heights reach almost twice as high. But they don't mention that in addition to dust, the dynamical properties are completely different as well. Stevens et al. (2019) showed for the exact same flight the vertical velocity measurements, which explain why there are

more clouds in one region than in the other.

For the same case study, the authors focus only on two single dropsondes. I would recommend to average over the dropsonde data from the single circle patterns to get more robust results. Otherwise, it comes across as cherry picking.

Minor comments:

Page 1, Line 4: "...impact on the Earth's radiation budget..." Shallow trade wind clouds have also an impact on the total precipitation in the tropics (Short and Nakamura, 2000), which has an important role for the boundary layer (Jensen et al., 2000)

Page 1, Line 5: The abbreviation is easier to understand when you change it to: "Next-generation Aircraft Remote-sensing for VALidation".

Page 1, Line 5: You mention the NARVAL studies, which include measurements from 2013, the NARVAL South campaign. In your manuscript, you only analyze data from the NARVAL2 campaign in 2016. By mentioning both campaigns in the abstract, it gives the impression that you analyze both data sets.

Page 1, Line 10 – 15: You mention the macro-physical properties of the clouds (shallower, lower cloud fraction) in dust-laden regions, but you don't explain why and how Saharan dust is causing that.

Page 1, Line 24 to page 2, Line 7: These are the only lines where you explain how SAL influence the cloud macro-physical properties. In addition to that, you should spend more time (before your conclusions) discussing these theories and connecting them to your results.

Page 2, Line 10 – 19: It is good to mention the prior campaigns, even if they didn't focus on trade wind clouds. Nevertheless, did these campaigns look also at the cloud macro-physical properties? If they did, are the results similar to your results?

Page 2, Line 26: You mention CloudSat here. The sensitivity of CloudSat is too low to

measure trade wind clouds properly. That might also be a reason why nobody looked at the interplay between SAL and trade wind clouds before by using these kind of data. The upcoming EarthCARE satellite might change that in the future.

Page 3, Figure 1: You show all flights in the figure, including the Ferry flights. Did you use the measurements during the ferry flights for your study? If this is the case, then you cannot call the studied clouds "trade wind clouds" anymore, because the ferry flights were outside of the trade wind region. Maybe you should only show the area and the flight paths, which you took for your analysis.

Page 3, Line 1 – 2: The sensitivity of the HAMP-radar is not high enough to detect shallow cumulus clouds. In comparison to the lidar measurements it should miss a lot of these smaller clouds. Do you use the radar in your study at all? It sounds like you do, because you say: " including radar and lidar systems - probably the two most important instruments for vertically highly resolved measurements of aerosol and cloud properties" . Maybe you can add a sentence to explain why you only analyze the lidar measurements.

Page 3, Line 3 – 5: Here you mention that your study is focusing only on NARVAL-II. The abstract gives the impression that you focus on NARVAL-South and NARVAL-II. The reader might wonder why you don't look into the campaign from 2013?

Page 3, Line 15: Replace "out" by "eastward".

Page 3, Line 16: "cruising altitude of ~15.5 km" Under empty (no crew and less fuel) conditions? What was the highest altitude during the campaign?

Page 3, Line 18: What does SMART stand for?

Page 4, Table 1: You mention the research objectives, but never mention the divergence flights, which were a big part of the campaign. For example, during flight 3 and 6 (see Stevens et al., 2019; Table 2). That's why the flight patterns show so many circles.

Page 4, Line 1: How much is "a large number of dropsondes"? During NARVAL-II 218 dropsondes were launched from HALO (see Stevens et al., 2019).

Page 5, Line 3: "resolution of approximately 200 m" is important for the identification of the cloud size. Later, you look at clouds with a cloud length smaller than 500 m (Fig. 6). Does it mean that these clouds consider only 2 pixels/data points?

Page 7, Figure 3: All your labels use a "/" to separate the Label text from the unit. Why not just writing the units in brackets "[°]", like you did it in Table 1 for UTC?

Page 7, Figure 3: You could use two different colors for the flight track to mark the dust and no dust regions.

Page 8, Line 5 – 6: Why were only RF4 and RF6 chosen to measure in dust free regions? Wouldn't it be useful to also take RF1, RF5 and RF7 to RF10 into account for measuring in dust free regions?

Page 8, Line 16: "far south" – How far?

Page 8, Line 24: Why are the trajectories not shown here? You could add another panel to Figure 3 and show the trajectories.

Page 8, Line 28: Maybe you can mark the dropsonde locations in Figure 3. Page 9, Figure 4: The upper left panel should be labeled as "D1", right?

Page 9, Figure 4: Why are you focusing only on two single dropsondes from each circle? It looks a little bit like cherry picking. I would suggest to average over all dropsondes for each circle.

Page 9, Line 6 – 7: You write "CF is 20 % in dust-free regions. In the SAL-region however, CF decreases to 11 % (including the clouds developing at the edges of the dust layer)", which gives the impression that only dust is causing the differences. You should clarify that.

Page 10, Line 1 – 2 : "However, in dust free regions cloud top heights reach almost

twice as high and up to 2 km." – This is not a direct effect from the dust. When you look at Figure 4 in Stevens et al. (2019), you see that the vertical velocities are different. Updrafts in the region of the NE circle cause the higher cloud top heights.

Page 11, Line 1 -2: Your conclusion that "less and shallower clouds" are present in "Saharan dust laden trade wind regions" is based only on Figure 5 and the CF. I think this conclusion need more fundamental explanation. For example, Figure 5 shows only a higher CTH fraction for dust laden regions in an altitude between 0.5 and 1.5 km. But why are there higher CTH fractions for dust free regions in all other altitude bins? The dust layer is in all flights between 2 and 6 km altitude. For this reason, you would also expect a higher CTH fraction in dust laden regions between 1.5 and 2 km. I am also wondering about the clouds between 0 and 0.5 km. Are these caused by precipitation? Usually, you would expect a cloud base height at ∼700m in the trade wind region.

Page 13, Line 8: Replace "flight" by "flights".

Page 13, Line 12: It looks like you can replace 1.6 km by 1.8 km.

Page 15, Line 5 – 7: You write before this sentence that the main player is a "dry anomaly". And then you say that "Saharan air layers were not found coming along with dry anomalies". For this reason, I think you cannot use this study to explain your results.

Page 15, Line 19 – 21: As a future study you could mention the upcoming EUREC4A campaign.

Page 19, Line 21: Replace "Klingenbiel" by "Klingebiel" Page 19, Line 22: Paper is already online and has a DOI.

References:

Jensen, J., S. Lee, P. Krummel, J. Katzfey, and D. Gogoasa, 2000: Precipitation in marine cumulus and stratocumulus. part I: Thermodynamic and dynamic observations of closed cell circulations and cumulus bands. Atmos. Res., 54, 117–155.

Short, D., and K. Nakamura, 2000: TRMM radar observations of shallow precipitation over the tropical oceans. J. Climate, 13, 4107–4124.

Stevens, B., F. Ament, S. Bony, S. Crewell, F. Ewald, S. Gross, A. Hansen, L. Hirsch, M. Jacob, T. Kölling, H. Konow, B. Mayer, M. Wendisch, M. Wirth, K. Wolf, S. Bakan, M. Bauer-Pfundstein, M. Brueck, J. Delanoë, A. Ehrlich, D. Farrell, M. Forde, F. Gödde, H. Grob, M. Hagen, E. Jäkel, F. Jansen, C. Klepp, M. Klingebiel, M. Mech, G. Peters, M. Rapp, A.A. Wing, and T. Zinner, 2019: A high-altitude long-range aircraft configured as a cloud observatory–the NARVAL expeditions. Bull. Amer. Meteor. Soc..

---

## Author Comment (AC1) · 13 Jun 2019

**Author's Comments**

Manuel Gutleben, Silke Groß and Martin Wirth

June 13, 2019

First of all, the authors would like to thank the referees for carefully reading the manuscript and for their helpful suggestions, feedbacks and comments. In the following, all comments and questions will be addressed and answered. The comments are repeated and a direct response is given below. In addition changes in the manuscript are highlighted by blue (additions) and red (removals) color and a marked-up manuscript version
5 (generated with latexdiff) is appended. We would like to start with the reply to Referee #2 due to thematic overlapping of several comments by both referees.

**Reply to Anonymous Referee #2**

**Major Comments concerning the structure**

**Comment:** The weakest part of this paper is the missing explanation about how the dust in the higher atmo-
10 sphere is influencing the cloud macro-physical properties. The authors mention some papers in the introduction and compare their results to some publications in the conclusions, but there is a section missing which explains the results in detail. For example, the authors mention a paper which shows a convection suppressing characteristic of the SAL with the main player being a dry anomaly in SAL-altitudes, but they cannot confirm a dry anomaly in SAL altitudes with their measurements. For this reason, the authors should make clear why they
15 still believe that SAL causes a suppressing characteristic on the clouds. In addition, the authors should make it very clear in the beginning (probably in the abstract or in the headline) that this study is a case study, which is based on four research flights.
**Response:** Thank you for this very helpful comment to improve the manuscript. In the revised manuscript we focused on answering all these questions/comments starting with an improved discussion of differences in
20 meteorological parameters in dust-laden compared to dust-free regimes. We compared our results to hypo-
thethes of previous studies that concentrated on the modification of boundary layer cloudiness due to enhanced boundary layer wind speeds and relative humidities to find possible other factors that play a role for convec-
tive suppression. However, wind speeds and moisture in both regimes do not differ significantly between both regimes. Thus, the elevated Saharan Air Layer seems to play the driving role in convective suppression and
25 may be used as a proxy for less and shallower clouds. We did not introduce a new chapter to address all these comments/questions and to relate previous findings to our results but rather discussed them in the respective chapters. Almost all made changes are also discussed in the following major and minor comments. Moreover, we agree with you that this study is a lidar case study and updated the title to "Cloud macro-physical properties in Saharan dust laden and dust free North Atlantic trade wind regimes: A lidar case study". All changes and
30 additions can be tracked in the appended marked-up manuscript version.

**Major Comments concerning the data set**

**Comment:** In this study, the authors focus on a dataset from the NARVAL-II campaign. In particular, they focus on four research flights where dust and dust free regions were sampled. If the authors would also look at
35 the measurements from the first NARVAL campaign in 2013, they could extend their dataset and improve the statistics.
**Response:** We followed your suggestion and extended our analysis to the dust-free winter season, by including sampled NARVAL-I data in trade wind regions to our analysis. We added NARVAL-I flight tracks to the shown map-plot (discussed in section *'Minor Comments'* of Referee #2) and included the cloud top height and cloud
40 (gap) length distribution to our analysis (shown below). This data set substantially extends the statistics and shows that higher-reaching boundary layer clouds embedded in a deeper marine boundary layer prevail during

winter season. Moreover, the cloud fraction was found to be higher in winter season and less short clouds but more short cloud gaps were detected. We discussed the derived distributions in the revised manuscript in the respective sections (see appended marked-up manuscript version).

[Figure]

Figure 1: Histograms of detected cloud top height fractions during NARVAL-I and II with bins of 0.5 km size. Red bars illustrate the distribution of cloud top height fractions in SAL-regions. Blue bars represent the derived cloud top height distribution from measurements in the dust-free trades during NARVAL-II. Grey bars show the derived cloud top height in the dust-free winter season during NARVAL-I.

[Figure]

Figure 2: Histograms of detected cloud lengths (top) and cloud gap lengths (bottom). Red bars illustrate the distribution of marine low cloud (gap) lengths located below Saharan dust layers. Blue bars represent the distribution derived from measurements in the dust-free trades during NARVAL-II. Grey bars show the derived distribution in the dust-free winter season during NARVAL-I.

**Major Comments concerning the analysis**

**Comment:** The authors compare the dust layer properties with the macro-physical properties of the clouds and see some correlations. However, the authors don't exclude other possibilities, which could cause the changes in clouds as well. For example, in their case study from the 19. August 2016, the authors compare a dust-laden region with a dust-free region and argue that in the latter, the cloud top heights reach almost twice as high. But they don't mention that in addition to dust, the dynamical properties are completely different as well. Stevens et al. (2019) showed for the exact same flight the vertical velocity measurements, which explain why there are more clouds in one region than in the other.

**Response:** Thank you for this valuable comment. In the revised manuscript, we discuss other possible factors causing the observed correlations in the respective regimes. As you already mentioned, Stevens et al. (2019) and Bony and Stevens (2019) show that derived vertical motions from dropsonde measurements indicate a rising motion in dust-free regions and a down-welling motion in dust-laden ones. These observations for sure explain why cloud tops are located in higher altitudes in dust-free regions. Nuijens et al. (2009) and Nuijens and Stevens et al. (2012) found that high wind speeds at surface level correspond to an increase of boundary layer humidity as well as vertical motion leading to a deepening of the cloud layer and increased rainfall. Lonitz et al. (2015) found from Large-Eddy simulations that even slight changes in boundary layer humidity impact shallow cloud development. We tested all these hypotheses by analyzing averaged vertical profiles of wind speed and direction as well as humidity from dropsonde measurements in the respective regimes (proposed in the next major comment and also addressed and suggested by Referee #1), but found no differences in boundary layer altitudes for these parameters. This is why we came to the conclusion that we believe that Saharan dust is changing meteorological conditions and can be used as a proxy for a decrease in cloud top height and cloud fraction. Earlier studies by Dunion and Velden (2004), Stephens et al. (2004) or Wong and Dessler (2005) fortify a suppressing characteristic of the Saharan Air Layer on convection analyzing passive remote sensing data sets. We addressed all these findings in our revised manuscript, related them to our findings an discussed them → see appended marked-up manuscript version)

**Comment:** For the same case study, the authors focus only on two single dropsondes. I would recommend to average over the dropsonde data from the single circle patterns to get more robust results. Otherwise, it comes across as cherry picking.

**Response:** To highlight the difference of measured lidar and dropsonde profiles in the respective regimes we made two changes in the revised manuscript. First, we plotted the flight track on top of MODIS real-color and and aerosol optical depth imagery, in a way that the two different regimes can easily be detected (see minor comments). Second, we followed your recommendation and averaged vertical profiles of all sondes in dust-laden and dust-free regimes. This makes the result more robust and shows that the previous profiles of single sondes were representative for both regimes. The main features persist in the revised graph. Following a suggestion of Referee #1 we have additionally plotted average profiles of wind speed and direction as well as humidity and added 4 plots which highlight the difference between the respective regimes. Moreover, we adapted the text to the reworked plot (not shown here - can bee seen in the appended marked-up manuscript version):

[Figure]

Figure 3: Mean vertical WALES lidar profiles of $\delta_{p532}$ and mean vertical dropsonde profiles of relative humidity ($r$), potential temperature ($\theta$), squared Brunt-Väisälä frequency $N^2 = \frac{g}{\Theta}\frac{d\Theta}{dz}$ as well as wind speed (u) and direction (wdir) in dust-free (I) and dust-laden (II) regions during RF6 on 19 Aug 2016 (horizontal bars indicate standard deviations). Differences in wind speed, potential temperature, relative humidity and water vapor mass mixing ratio (MR) between the two regions are shown in (III).

**Minor Comments**

**Comment:** Page 1, Line 4: ...impact on the Earths radiation budget... Shallow trade wind clouds have also an impact on the total precipitation in the tropics (Short and Nakamura, 2000), which has an important role for the boundary layer (Jensen et al., 2000).

**Response:** We modified the sentence in the abstract to address these points: "Shallow trade wind clouds have a significant impact on the Earth's radiation budget as well as on total tropical precipitation and marine boundary layer dynamics and are still introducing large uncertainties in climate sensitivity estimates, because of their poor representation in climate models."

**Comment:** Page 1, Line 5: The abbreviation is easier to understand when you change it to: "Next generation Aircraft Remote-sensing for VALidation".

**Response:** We followed the suggestion and changed the explanation of the abbreviation in the text.

**Comment:** Page 1, Line 5: You mention the NARVAL studies, which include measurements from 2013, the NARVAL South campaign. In your manuscript, you only analyze data from the NARVAL2 campaign in 2016. By mentioning both campaigns in the abstract, it gives the impression that you analyze both data sets.

**Response:** We clarified that by adding the NARVAL-I data set to the analysis (explained in detail in section *'Major comments concerning the data set'*).

**Comment:** Page 1, Line 10 15: You mention the macro-physical properties of the clouds (shallower, lower cloud fraction) in dust-laden regions, but you don't explain why and how Saharan dust is causing that.

**Response:** We added a sentence which sums up the major findings of our conducted dropsonde analysis to addresses the presence of two additional inversions which are stabilizing the environment and are counteracting convective development: The cloud fraction in the dust-laden summer trades is only 14 % compared to a fraction of 31 % and 37 % in dust-free trades and the winter season. Dropsonde-measurements show that long-range transported Saharan dust layers come along with two additional inversions which counteract convective

development, stabilize the stratification and may lead to a decrease of convection in those areas."

**Comment:** Page 1, Line 24 to page 2, Line 7: These are the only lines where you explain how SAL influence the cloud macro-physical properties. In addition to that, you should spend more time (before your conclusions) discussing these theories and connecting them to your results.
**Response:** Within the revised manuscript we explained all those theories and discussed them in the respective sections. As also proposed by Referee #1 we added a paragraph in the Introduction and elaborate in more detail how dust impacts cloud-formation and why dust can act as a good cloud condensation nucleus. Furthermore, we discussed hypothethes by Nuijens et al. (2009), Nuijens and Stevens (2012) and Lonitz et al. (2015) (discussed in section "*Major comments*" and also addressed and suggested by Referee #1) that changes in boundary layer wind speed and humidity modify vertical motion and thus cloud development and related them to our results. In the revised dropsonde analysis we discuss changes in SAL-associated meteorology and that additional SAL-related inversions counteract convective development (see appended marked-up manuscript version).

**Comment:** Page 2, Line 1019: It is good to mention the prior campaigns, even if they didn't focus on trade wind clouds. Nevertheless, did these campaigns look also at the cloud macro-physical properties? If they did, are the results similar to your results?
**Response:** To our knowledge none of the mentioned campaigns investigated macro-physical cloud properties. Those campaigns rather focused on aerosol and cloud micro-physics to study the aerosol and cloud radiative effect: "Within this series of closure experiments, which included airborne and ground-based in-situ and remote sensing measurements as well as modeling efforts, micro-physical, chemical and radiative properties of dust were investigated at the beginning of its long-range transport near the source regions as well as after its long-range transport in the vicinity of Barbados."

**Comment:** Page 2, Line 26: You mention CloudSat here. The sensitivity of CloudSat is too low to measure trade wind clouds properly. That might also be a reason why nobody looked at the interplay between SAL and trade wind clouds before by using these kind of data. The upcoming EarthCARE satellite might change that in the future.
**Response:** Thank you for your comment. We added some words addressing the resolution of CloudSat and the future EarthCARE mission: "Satellites with an active remote sensing payload, e.g. the Cloud-Aerosol Lidar and Infrared Pathfinder Satellite Observation (CALIPSO; Winker et al., 2010) and CloudSat (Stephens et al., 2002) provide vertically highly resolved measurements of aerosol and cloud properties with nearly global coverage (Liu et al., 2008; Medeiros et al., 2010). Up to now, studies based on active remote-sensing satellite data with focus on cloud macro-physical properties concentrated on long-term and large-scale observations, e.g. low-latitude boundary layer cloud cover (Medeiros et al., 2010), as the sensitivity of those instruments is too low to detect shallow marine clouds with high resolution. The upcoming EarthCARE (Earth Clouds, Aerosols and Radiation Explorer) satellite mission which is planned to be launched in 2021 (Illingworth et al., 2015) might change that in future due to its unique payload: a combination of lidar (Atmospheric Lidar - ATLID) and Cloud Profiling Radar (CPR)."

**Comment:** Page 3, Figure 1: You show all flights in the figure, including the Ferry flights. Did you use the measurements during the ferry flights for your study? If this is the case, then you cannot call the studied clouds "trade wind clouds" anymore, because the ferry flights were outside of the trade wind region. Maybe you should only show the area and the flight paths, which you took for your analysis.
**Response:** We clarified this in the revised manuscript and gave an explaination. Furthermore, we revised the map-plot showing HALO-flight tracks. We only plotted flight tracks used in this study and zoomed into the trade-wind region. This makes it much clearer for the reader: "Data-sets obtained during the NARVAL-II transfer flights from and to Germany (i.e. RF1 30 and RF10) are not included in the analysis, because most measurements took place outside the trades and cirrus fields were present inside the trades. RF5 and 7 are excluded as well since cirrus fields covered most of the research area during RF5 and RF7's objective was to cross the Inter Tropical Convergence Zone (ITCZ) for several times. NARVAL-I lidar data (obtained from measurements inside the trades (10 to 20° N)) are used to compare obtained results from the 2016 summer season to the 2013 winter season."

[Figure]

Figure 4: NARVAL research flight tracks: NARVAL-II dust-flights (color coded), NARVAL-II dust-free flights (light-grey), NARVAL-I (dark-grey).

**Comment:** Page 3, Line 1-2: The sensitivity of the HAMP-radar is not high enough to detect shallow cumulus clouds. In comparison to the lidar measurements it should miss a lot of these smaller clouds. Do you use the radar in your study at all? It sounds like you do, because you say: including radar and lidar systems - probably the two most important instruments for vertically highly resolved measurements of aerosol and cloud properties. Maybe you can add a sentence to explain why you only analyze the lidar measurements.

**Response:** We do not use radar data in the analysis and modified the paragraph as follows: "For this purpose it was equipped with a combined active and passive remote sensing payload, including a radar and a lidar system... The sensitivity of the radar system is not high enough to detect small-scale shallow cumulus clouds as well as aerosol layers. This is why this study only focuses on the retrieval of horizontal and vertical distributions of both aerosols and clouds from lidar measurements performed during the NARVAL field campaigns to study the impact of the SAL on subjacent marine cloud macro-physical properties (i.e. cloud fraction, cloud top height, cloud length)."

**Comment:** Page 3, Line 3-5: Here you mention that your study is focusing only on NARVAL-II. The abstract gives the impression that you focus on NARVAL-South and NARVAL-II. The reader might wonder why you don't look into the campaign from 2013?

**Response:** We clarified that by adding the NARVAL-I data set to the analysis (explained in detail in section *'Major comments concerning the data set'*).

**Comment:** Page 3, Line 15: Replace "out" by "eastward".

**Response:** We replaced the word: "...HALO was operated eastward of Barbados."

**Comment:** Page 3, Line 16: "cruising altitude of 15.5 km" Under empty (no crew and less fuel) conditions? What was the highest altitude during the campaign?

**Response:** According to the official webpage, HALO has a certified ceiling of 15.545 km altitude. We also added the highest altitudes during the NARVAL campaign series: "The aircraft has a maximum range of more than 12000 km and certified ceiling of 15.545 km altitude (max altitudes: NARVAL-I: ~14 km; NARVAL-II: ~15 km)."

**Comment:** Page 3, Line 18: What does SMART stand for?

**Response:** It stands for "Spectral Modular Airborne Radiation measurement sysTem". We added this information to the revised manuscript.

**Comment:** Page 4, Table 1: You mention the research objectives, but never mention the divergence flights, which were a big part of the campaign. For example, during flight 3 and 6 (see Stevens et al., 2019; Table 2). Thats why the flight patterns show so many circles.

**Response:** Of course divergence flights were a big part of the campaign. We added two sentences to address this point: "Moreover, studying the large scale atmospheric divergence was a main objective of the campaign

(Bony and Stevens, 2019). This is why the flight patterns show many circles, i.e during RF2, 3, 6-8 and 10."

**Comment:** Page 4, Line 1: How much is "a large number of dropsondes"? During NARVAL-II 218 dropsondes were launched from HALO (see Stevens et al., 2019).

**Response:** We added the number of deployed sondes "Additionally a large number of dropsondes were deployed to get information on the atmospheric state (NARVAL-I: 71; NARVAL-II: 218)."

**Comment:** Page 5, Line 3: "resolution of approximately 200 m" is important for the identification of the cloud size. Later, you look at clouds with a cloud length smaller than 500 m (Fig. 6). Does it mean that these clouds consider only 2 pixels/data points?

**Response:** We clarified the sampling frequency of WALES in the chapter *"The WALES instrument"* and show that small clouds are detected on the basis of averaged measurements with even higher resolution: "The temporal resolution of the raw data is 5 Hz and is averaged to 1 Hz for a better signal-to noise ratio. This results in a horizontal resolution of approximately 200 m at typical aircraft speed."

**Comment:** Page 7, Figure 3: All your labels use a "/" to separate the Label text from the unit. Why not just writing the units in brackets "[°]", like you did it in Table 1 for UTC?

**Response:** We changed the labeling of all graphs from slashes to brackets.

**Comment:** Page 7, Figure 3: You could use two different colors for the flight track to mark the dust and no dust regions.

**Response:** To highlight dust-laden and dust-free regions we plotted the flight track on top of MODIS real-color and and aerosol optical depth imagery. The two different regimes can now be easily distinguished by the reader. Additionally we color-coded launched dropsondes: sondes in dust-free regimes (blue); sondes in dust-laden regimes (red). The new figure is described in the revised manuscript as follows: "Whereas the first pair of circles was performed over a heavily dust-laden region in the southern part of the flight track, the second pair was performed in the northern part over an almost dust-free region. This is also seen in MODIS aerosol optical depth imagery at 13:40 UTC in Figure 4 (right) where the region around the southern circle shows a maximum aerosol optical depth greater 0.4."

[Figure]

Figure 5: Flighttrack of RF6 on 19 Aug 2016 on top of the Terra-MODIS (MODerate-resolution Imaging Spectroradiometer) true color image (left) and the MODIS aerosol optical depth (AOD) product (right) at 13:40 UTC. Launched dropsondes are marked by colored dots (red dots: mineral dust laden regions, blue: dust free regions).

**Comment:** Page 8, Line 5 6: Why were only RF4 and RF6 chosen to measure in dust free regions? Wouldn't it be useful to also take RF1, RF5 and RF7 to RF10 into account for measuring in dust free regions?

**Response:** We clarified this in the revised manuscript and gave an explaination: Data-sets obtained during the NARVAL-II transfer flights from and to Germany (i.e. RF1 30 and RF10) are not included in the analysis, because most measurements took place outside the trades and cirrus fields were present inside the trades. RF5

and 7 are excluded as well since cirrus fields covered most of the research area during RF5 and RF7's objective was to cross the Inter Tropical Convergence Zone (ITCZ) for several times.

**Comment:** Page 8, Line 16: "far south" How far?

**Response:** We added this information as follows: The Intertropical Convergence Zone (ITCZ) and associated deep convection were located ~550 km south of the flight track at around 10°N and it is not expected to have an influence of the ITCZ on our analysis.

**Comment:** Page 8, Line 24: Why are the trajectories not shown here? You could add another panel to Figure 3 and show the trajectories.

**Response:** We followed your suggestion and added a figure which shows the calculated trajectories for all four research flights leading over Saharan dust layers on a map as well as a function of time and altitude. Moreover, we addressed the new figure in the revised manuscript: The Saharan origin of the observed dust layers is verified using 10-day backward-trajectories with starting points at the center of the respective Saharan air layers (Figure 3). All observed dust layers traveled for 5 to 10 days from the Adrar-Hoggar-Aïr region to the measurement location over the western North Atlantic Ocean. In central Africa the SAL is formed by intense surface heating and dry convection which mixes dust particles to altitudes of up to 6 km (Gamo, 1996).

[Figure]

Figure 6: 10-day backward trajectories with starting points at the center of the respective Saharan air layers for the four NARVAL-II research flights leading over Saharan dust-laden trade wind regions (RF2, 3, 4 and 6).

**Comment:** Page 8, Line 28: Maybe you can mark the dropsonde locations in Figure 3. Page 9, Figure 4: The upper left panel should be labeled as "D1", right?

**Comment:** Page 9, Figure 4: Why are you focusing only on two single dropsondes from each circle? It looks a little bit like cherry picking. I would suggest to average over all dropsondes for each circle.

**Response to both comments:** We averaged all sampled dropsonde-profiles in dust-laden and dust-free regions (explained in detail in section *'Major comments concerning the analysis'*), marked dropsonde locations on top of MODIS real-color as well as aerosol-optical depth imagery (shown above) and revised the mentioned figure.

**Comment:** Page 9, Line 67: You write "CF is 20% in dust-free regions. In the SAL-region however, CF decreases to 11% (including the clouds developing at the edges of the dust layer)", which gives the impression that only dust is causing the differences. You should clarify that.

**Comment:** Page 10, Line 12 : "However, in dust free regions cloud top heights reach almost twice as high and up to 2 km." This is not a direct effect from the dust. When you look at Figure 4 in Stevens et al. (2019), you see that the vertical velocities are different. Updrafts in the region of the NE circle cause the higher cloud top heights.

**Response to both comments:** Within the revised manuscript we highlight the differences in dynamics by discussing the major findings of the recently published paper by Bony and Stevens (2019) and Stevens et al. (2019) and relating regions of up-welling and down-welling vertical motion to dust-free and dust-laden regimes. Additionally, we give an revised, extended and more profound discussion of differences in meteorological parameters in the two regimes in section 3.2. (addressed in section *'Major comments'* of Referee #2): However, in dust free regions cloud top heights reach almost twice as high and up to 2 km. Divergence measurements discussed by Bony and Stevens (2019) and Stevens et al. (2019) show that dynamical properties in the two regions are different as well. They found that MBL-vertical velocity in the dust-free regime is directed upwards and could explain the observed increased cloud top heights. For an investigation of the question why vertical wind speeds, cloud tops and cloud fractions are higher in the dust-free regime than in the SAL-regime differences in meteorological parameters between SAL-regions and dust-free regions are analyzed by discussing mean profiles of all dropsonde measurements in the respective regions.

**Comment:** Page 11, Line 1 -2: Your conclusion that "less and shallower clouds" are present in Saharan dust laden trade wind regions" is based only on Figure 5 and the CF. I think this conclusion need more fundamental explanation. For example, Figure 5 shows only a higher CTH fraction for dust laden regions in an altitude between 0.5 and 1.5 km. But why are there higher CTH fractions for dust free regions in all other altitude bins? The dust layer is in all flights between 2 and 6 km altitude. For this reason, you would also expect a higher CTH fraction in dust laden regions between 1.5 and 2 km. I am also wondering about the clouds between 0 and 0.5 km. Are these caused by precipitation? Usually, you would expect a cloud base height at ∼700m in the trade wind region.

**Response:** Figure 5 shows the cloud top height fractions in the respective regimes as a function of relative frequency. We think this comment is probably a misunderstanding, since both shown distributions are independent from each other and just show that cloud tops detected in dust-free regions show a broader distribution and reach higher up into the atmosphere than cloud tops in dust-laden regions. Those cloud tops were never observed to reach higher than 2.5 km altitude during the whole field campaign. As an example, more than 60% of all detected cloud tops in dust-laden regimes (not of the total amount of observed cloud tops in both regimes) were observed in the interval from 0.5 to 1 km altitude.

Furthermore, we added two sentences that explain the existence of cloud top heights in the lowermost 500 m. In our opinion those low CTHs are not representing precipitation, because precipitating clouds in general are deeper: Several clouds were also detected in the lowermost 0.5 km of the atmosphere (∼1 %). Most likely those clouds are evolving or dissipating clouds at the bottom of the cloud layer."

**Comment:** Page 13, Line 8: Replace "flight" by "flights".
**Response:** We replaced the word.

**Comment:** Page 13, Line 12: It looks like you can replace 1.6 km by 1.8 km.
**Response:** Thank you for the comment. We corrected the number.

**Comment:** Page 15, Line 5  7: You write before this sentence that the main player is a "dry anomaly". And then you say that "Saharan air layers were not found coming along with dry anomalies". For this reason, I think you cannot use this study to explain your results.
**Response:** We clarified that and made clear that we cannot explain our found results with findings of this specific study. We highlighted that we found enhanced concentrations of water vapor in SAL-altitudes compared to the dry free troposphere (which has already been observed, e.g. by Jung et al. (2013)) and that we also find a suppressing characteristic of the SAL on convection: "These results are in good agreement with results of previous satellite remote sensing studies (Dunion and Velden, 2004) and model studies (Wong and Dessler, 2005; Stephens et al., 2004) which also suggest a convection-suppressing characteristic of the SAL. Some of those studies suggest that the main player of the suppression characteristic is a dry anomaly in SAL-altitudes. However, all observed long-range transported Saharan air layers during NARVAL-II were not found to come along with dry anomalies, but were rather showing enhanced humidities (compared to the surrounding dry free trade wind atmosphere) in the range from 2 to 4 $gkg^1$. Saharan air layers frequently show water vapor mixing ratios in this range over Africa (Marsham et al., 2008). During the transport towards the Caribbean the SAL conserves the received moisture and takes up additional one from upwelling surface fluxes during transport (Jung et al., 2013). Nevertheless, a suppressing characteristic of the SAL on subjacent marine clouds is evident as well."

**Comment:** Page 15, Line 19  21: As a future study you could mention the upcoming EUREC4A campaign.

**Response:** We followed your suggestion and mentioned EUREC$^4$A in section *'Summary and Conclusion'*: "Further reaching questions regarding changes in radiation caused by the dust layer and its moisture, changes in the general circulation patterns or the settling of dust particles into the cloud layer (Gross et al., 2016) could not be addressed within the present work and are left to future studies and field campaigns, e.g. the upcoming EUREC$^4$A field campaign (ElUcidating the RolE of Clouds-Circulation Coupling in ClimAte) in early 2020 (Bony et al., 2017)."

**Comment:** Page 19, Line 21: Replace "Klingenbiel" by "Klingebiel" Page 19, Line 22: Paper is already online and has a DOI.

**Response:** We rectified the transposed combination of letters and added the DOI.

**Reply to Anonymous Referee #1**

**General Comments**

**Comment:** Reading through the paper suggests that the aerosols are causing the differences in cloudiness but how can an elevated dust layer imprint itself on the lower laying clouds? Furthermore, what is not quite clear to me is how the meteorology during periods of SAL differs to periods without dust and how this might influence the cloudiness. Other studies (e.g. Lonitz et. al 2015) have shown, that e.g. slight differences in relative humidity can also cause differences in trade wind clouds to a comparable level as differences in aerosol load can do. Also effects of wind should be discussed, as done by Nuijens and Stevens in 2012 using Large Eddy Simulations. Having access to meteorological data from the dropsondes, I would like to see that the differences in relative humidity and wind are analysed in greater depth (not just shown as done in Fig. 4 for humidity) and a discussion about if dust or the meteorology associated with dust could potentially can cause the observed changes in cloudiness.

**Response:** We follow your suggestion and analyzed dropsonde data in greater depth. In the revised manuscript we averaged profiles from all dropsondes launched in dust-laden and dust-free regimes and discussed and explained measured differences (see major comments of Referee #2). We discussed hypothethes how boundary layer wind speed and humidity could cause changes in marine boundary layer cloudiness (as shown in Nuijens et al. (2009), Nuijens and Stevens et al. (2012) and Lonitz et al. (2015)) and found that there is no significant difference in surface wind speed and boundary layer humidity between both regimes, which could explain upwelling vertical motion and cloud development. Furthermore, we highlighted changes in meteorology (i.e. additional inversions) that come along with an elevated dust layer, which could potentially explain the observed changes in cloudiness. Summed up, the SAL potentially modifies atmospheric stability and radiative transfer and appears to have a suppressing impact on convection below. All these changes and additions are marked up in the appended revised manuscript.

**Specific Comments**

**Comment:** Introduction: Please elaborate in more detail how dust and the formation of cloud interact, how the aerosol could alter clouds and why dust can act as a good CCN.

**Response:** In the revised manuscript we described how dust and the formation of cloud may interact, how aerosol could alter clouds and why dust can act as a good CCN: "During its long-range transport the SAL affects the Earth's radiation budget in two different ways. First, mineral dust aerosols may act as cloud condensation nuclei (CCN) or ice nucleating particles (INPs) in water and ice clouds, hence influencing cloud microphysics - this effect is referred to as 'indirect' mineral dust aerosol effect (Twomey, 1974, 1977; Karydis et al., 2011; Begué et al., 2015; DeMott et al., 2015; Boose et al., 2016). Thus, cloud formation, lifetime and occurrence as well as precipitation and ice formation may be manipulated by Saharan dust deposition into the cloud layer (Mahowald and Kiehl, 2003; Seifert et al. 2010). Secondly, dust particles absorb and scatter solar radiation during daytime and emit thermal radiation during nighttime. This so-called 'direct' mineral dust radiative effect modifies the atmospheric temperature profile and impacts the evolution of atmospheric stratification, sea surface temperature as well as cloud development (Carlson and Benjamin, 1980; Lau and Kim, 2007)."

**Comment:** p2, l18/19 is not true. Lonitz et al 2015 (https://journals.ametsoc.org/doi/10.1175/JASD-14-0348.1) have studied this.

**Response:** Thank you very much for your reference to Lonitz et al. (2015). We have read their paper and used it as a reference for the Introduction: "Although the interaction of Saharan dust layers and clouds has already been a focus during these campaigns and other studies, e.g. by investigating glaciation of mixed-phase clouds (Ansmann et al, 2008; Seifert et al., 2010) or by exploring the relationship between shallow cumulus precipitation rates and radar measurements in dust-laden and dust-free environments (Lonitz et al., 2015), the impact of long-range transported elevated Saharan dust on cloud macro-physical properties of subjacent trade wind clouds has not been studied."

**Comment:** Section 2.4: Do you also check if the neighboring cloudy profiles have the cloud in similar height levels? If not, this might have an impact on the results shown in section 3.3.

**Response:** We checked whether this is the case. However, almost every cloud shows cloud tops in similar altitudes and due to the large overall number of detected clouds no impact on derived statistics is seen.

**Comment:** Figure 4 and p15, l 5-7: Was the relative humidity measured by the dropsondes or derived by the lidar? How does dry Saharan air layers relate to an increase in humidity?

**Response:** Moisture profiles were measured by both dropsondes and the lidar instrument and agree within a few percent, which is in the order of measurement uncertainties (Stevens et al., 2017).

In the revised manuscript, we addressed that long-range transported Saharan Air Layers are known to show an increased amount of water vapor compared to the dry free troposphere: "Moreover, enhanced amounts of water vapor are observed inside the long-range transported SAL. Relative humidity and water vapor mass mixing ratio show an increase of $2\,\mathrm{g\,kg^{-1}}$ in SAL regions compared to the dust-free trade wind region. Such an increase has already been observed by (Jung et al., 2013). From radiosonde measurements they found that the SAL transports moisture from Africa towards the Caribbean and gets moistened during transport by upwelling surface fluxes."

Citation: *Stevens, B., Brogniez, H., Kiemle, C., Lacour, J.-L., Crevoisier, C., and Kiliani, J.: Structure and Dynamical Influence of Water Vapor in the Lower Tropical Troposphere, Surv. Geophys., 38, 13711397, https://doi.org 10.1007/s10712-017-9420-8, 2017.*

**Comment:** P15, l14/15: How does the suppression work. Please elaborate.

**Response:** We deleted the sentences and added some sentences that explain how optically and vertically thicker dust layers could potentially suppress the evolution of higher reaching convection: "These results indicate that optically and vertically thick elevated Saharan dust layers have a greater suppressing effect on convection below than optically and vertically thin layers. Dropsonde profiles of potential temperature $\Theta$ and the squared Brunt-Väisälä frequency $N^2$ in dust-laden trade wind regions indicate two inversions at the bottom and the top of the SAL which additionally counteract convective development. Those two SAL-related additional inversion layers are an explanation why there are less and shallower clouds in SAL regions and why thick and pronounced dust layers introduce a more stable stratification to the trade wind regime than less pronounced ones." Additionally we discussed hypothethes on how elevated and long-range transported Saharan dust layers and the associated change in meteorological parameters can have an impact on cloud-development below throughout the revised manuscript (discussed in author comments above).

**Technical Comments**

**Comment:** Often the word "underneath" is used instead of "below", e.g. p2 l8, caption of figure 7.

**Response:** We replaced the word "underneath" with "below" throughout the text.

**Comment:** P2, l35: Citation incorrect: Stevens et al 2019.

**Response:** The citation is updated in the revised manuscript. Stevens et al. (2019) is already published and not subject of the review-process anymore.

**Comment:** P10, l19: "first" instead of "fist"

**Response:** We corrected that.

**Comment:** Sec 3.3.2: How do you derive the cloud length in kilometers? My guess is that you assume some relationship between the speed of the aircraft and time?!

**Response:** Cloud (gap) lengths are calculated using aircraft geolocation and mean cloud top height using the haversine formula. We clarified that in the revised manuscript: For the calculation of cloud lengths along the flight path neighboring cloud-flagged vertical profiles are connected. The cloud length is then determined as a function of the respective geolocations (aircraft latitude and longitude) and CTH using the haversine formula.

**Besides changes which were made by reason of referee comments, some small textual and grammatical corrections were made during proofreading, with the most prominent one being an additional column in Table 1 showing the total duration of the research flights. All made changes can be found in the attached marked up manuscript version.**

[revised manuscript text omitted]

---

## Referee Report (RR1)

General comments:

This version of the paper investigating the link between the Saharan aerosol layer and cloudiness using lidar measurements and dropsondes data of four NARVAL flights has been much improved by the authors. The analysis has been extended by looking into vertical profiles of potential temperature, humidity, wind, etc. from dropsondes, checking conditions during NARVAL I and discussing findings with what has been found by others. In particular I welcome the analysis around the reason why the presence of SAL is correlated well with the occurrence of clouds and their macro-physical properties. It has been found that many meteorological properties between dusty and non-dusty times are quite similar with the exception of the relative humidity inside SAL or for example the number of inversions present. It has been highlighted that more work needs to be done to fully understand if those differences in cloudiness are a consequence of radiative effects by the dust, by dust settlement or by changes in the general circulation patterns. I recommend this paper for publications after some final minor editing.

Specific comments:

The main body of the manuscript has been much improved. However, the abstract does not reflect this nice work. I would like to see more findings presented instead of introduction. Additionally, I think that the abstract has to be a bit more concise regarding the effect of dust. There are two effects dust can have; either directly or indirectly by modifying the cloud properties or the air properties associated with the dust layer (SAL) can suppress cloud formation (not the dust per se). Those are two different things and need to be articulate better, especially because the abstract begins talking about direct and indirect effects of aerosol but finishes with the effect of SAL. In the summary that has been done much better by discussing that the real reason behind the correlation of SAL and cloud properties is still open for research.

P2, l6: Dust is known to be not a good CCN! Only when it accumulates soluble material through internal mixing it can act as CCN. Please add this information. It is a good IN though.

Technical comments:

P5, l11:  Change "RF5  and 7  are excluded as well since cirrus fields covered most of the research area during RF5  and RF7 's objective was to cross the Inter Tropical Convergence Zone (ITCZ) for several times." to  "RF5  and RF7  are also excluded because cirrus fields covered most of the research area during RF5  and the objective of RF7 was to cross the Inter Tropical Convergence Zone (ITCZ) for several times.".

P5,l14: Please do not use ")".

P5,l15: Change "Altogether a 22 h -lidar data set measured in the dust-free trades, a 16 h - lidar data set measured in SAL trade wind regions and a 44 h -data set obtained in winter season is used to study differences in macro-physical cloud properties between the respective regions and seasons." To "In summary, 38 hours of measurements during the

summer season (22 hours of lidar measurements during dust-free times, 16 hours of lidar measurements with SAL present) and 44 hours of measurements during the winter season are used to study differences in macro-physical cloud properties between the dust and non-dusty times and different seasons.".

P5,l33: Add ", respectively" in the end.

P11,l23/24: Remove "so-called" and change to "a" and change "from ocean surface to 0.5 to 0.7km and the cloud " to "from the ocean surface to 0.5 -  0.7km and a cloud".

P17,l11/11: Change "It can be summarized that dust-laden regions implicate less, shallower and smaller clouds than dust-free regions." to "It can be summarized that during SAL less, shallower and smaller clouds are present than during times without SAL.".

---

## Author Response (AR2)

**Author's Reply to the discussion of the revised manuscript version**

Manuel Gutleben, Silke Groß and Martin Wirth

July 12, 2019

The authors would like to thank the referees for carefully reading the revised manuscript for a second time and for their helpful final suggestions, feedbacks and comments. In the following, all remaining minor comments and questions will be addressed and answered. The comments are repeated and a direct response is given below. In addition changes in the manuscript are highlighted in the appended marked-up manuscript version blue (additions) and red (removals) color.

**Reply to Minor Comments of Anonymous Referee #1**

**Specific comments**

*Comment: The main body of the manuscript has been much improved. However, the abstract does not reflect this nice work. I would like to see more findings presented instead of introduction. Additionally, I think that the abstract has to be a bit more concise regarding the effect of dust. There are two effects dust can have; either directly or indirectly by modifying the cloud properties or the air properties associated with the dust layer (SAL) can suppress cloud formation (not the dust per se). Those are two different things and need to be articulate better, especially because the abstract begins talking about direct and indirect effects of aerosol but finishes with the effect of SAL. In the summary that has been done much better by discussing that the real reason behind the correlation of SAL and cloud properties is still open for research.*

Thank you for the valuable comment. We agree that the abstract includes too many introductory sentences and is not concise regarding the SAL-effects on marine clouds. This is why we eliminated most of the introductory part and focused on summing up our findings. In our opinion the new and revised abstract (which can be found in the marked-up manuscript version) is concise and coherent and sums up the results of the study.

*Comment: P2, l6: Dust is known to be not a good CCN! Only when it accumulates soluble material through internal mixing it can act as CCN. Please add this information. It is a good IN though.*

We added this information.

**Technical comments**

*Comment: P5, l11: Change "RF5 and 7 are excluded as well since cirrus fields covered most of the research area during RF5 and RF7 s objective was to cross the Inter Tropical Convergence Zone (ITCZ) for several times." to "RF5 and RF7 are also excluded because cirrus fields covered most of the research area during RF5 and the objective of RF7 was to cross the Inter Tropical Convergence Zone (ITCZ) for several times.".*

We changed the sentence.

*Comment: P5, l14: Please do not use "))".*

We changed that.

*Comment: P5, l15: Change "Altogether a 22 h -lidar data set measured in the dust-free trades, a 16 h - lidar data set measured in SAL trade wind regions and a 44 h -data set obtained in winter season is used to study differences in macro-physical cloud properties between the respective regions and seasons." to "In summary, 38 hours of measurements during the summer season (22 hours of lidar measurements during dust-free times, 16*

*hours of lidar measurements with SAL present) and 44 hours of measurements during the winter season are used to study differences in macro-physical cloud properties between the dust and non dusty times and different seasons.".*

We followed your suggestion and made the change.

*Comment: P5,l33: Add ", respectively" in the end.*

Done.

*Comment: P11,l23/24: Remove "so-called" and change to "a" and change "from ocean surface to 0.5 to 0.7km and the cloud" to "from the ocean surface to 0.5 - 0.7km and a cloud".*

We did that.

*Comment:P17,l11/11: Change "It can be summarized that dust-laden regions implicate less, shallower and smaller clouds than dust-free regions." to "It can be summarized that during SAL less, shallower and smaller clouds are present than during times without SAL.".*

We followed your suggestion and modified the sentence.

**Reply to Minor Comments of Anonymous Referee #1**

*Comment: Page 1, Line 9-12: I would suggest removing the sentence "These observations...in the dust-free winter-season". Instead, you can change the sentence before to: "Airborne lidar measurements in the vicinity of Barbados performed for both campaigns are used to investigate...Saharan dust layers."*

We modified those sentences according to your suggestion.

*Comment: Page 4, Text for Figure 1: NARVAL-1 was also dust free, right? Maybe you can write then: NARVAL-I and NARVAL-II dust-free flights (light grey). I had a hard time seeing the differences between the dark grey, green and red track on my printed copy. Maybe you can just use the light grey color for the dust-free flights. It doesn't really matter if it was a flight from the first or the second NARVAL campaign.*

Thank you for your comment on this figure. We changed the coloring and use grey color for dust-free flights during both campaigns in the revised plot. Moreover, we slightly changed the coloring of the flight tracks in dust-laden regions to help people with with red-green visual impairments.

*Comment: Page 5, Line 11: Be consistent with the following line and write "RF7" instead of just "7".*

We changed that.

*Comment: Page 8, Line 3 6: Maybe it makes sense to show Figure 3 before Figure 2 because you explain Figure 3 in the first paragraph of subsection 3.1.*

We exchanged the figures.

*Comment: Page 10, Figure 5: 1. Exchange the lower left plot (Alt vs delta u) with the lower right plot (delta MR). Then you have all velocities in one column. 2. Add a letter (a,b,c,) to every plot and refer to it in the text. It makes it much easier to find the plot that you are talking about. It also makes it easier to find the associated explanation in the text.*

We exchanged the plots, added letters and referred to the plots in the text.

*Comment: Page 13, Figure 6: The label of the vertical axis is wrong. It should be replaced by "Altitude [m]".*

Thank you for pointing this out. We corrected the axis-labeling.

*Comment: Page 13, Line 16: The sentence "In both regions the frequency of cloud length occurrence decreases strongly with increasing cloud length." is not true for a cloud length $> 5$. There you see an increase.*

This is of course true. We clarified that.

*Comment: Page 15, Figure 8: The label in the lower right plot shows a "T" instead of a "Tau". Maybe you should also add letters here, like I suggested for Figure 5.*

We corrected the labeling and added letters to the figure.

[revised manuscript text omitted]